# Data pruning and neural scaling laws: fundamental limitations of score-based algorithms

**Fadhel Ayed**[*]                                                             *fadhel.ayed@gmail.com*
*Huawei Technologies France*

**Soufiane Hayou**[*]                                                          *hayou@nus.edu.sg*
*National University of Singapore*

**Reviewed on OpenReview:** *https://openreview.net/forum?id=iRTL4pDavo*

## Abstract

Data pruning algorithms are commonly used to reduce the memory and computational cost of the optimization process. Recent empirical results (Guo, B. Zhao, and Bai, 2022) reveal that random data pruning remains a strong baseline and outperforms most existing data pruning methods in the high compression regime, i.e. where a fraction of 30% or less of the data is kept. This regime has recently attracted a lot of interest as a result of the role of data pruning in improving the so-called neural scaling laws; see (Sorscher et al., 2022), where the authors showed the need for high-quality data pruning algorithms in order to beat the sample power law. In this work, we focus on score-based data pruning algorithms and show theoretically and empirically why such algorithms fail in the high compression regime. We demonstrate "No Free Lunch" theorems for data pruning and discuss potential solutions to these limitations.

## 1 Introduction

Coreset selection, also known as data pruning, refers to a collection of algorithms that aim to efficiently select a subset from a given dataset. The goal of data pruning is to identify a small yet representative sample of the data that accurately reflects the characteristics of the entire dataset. Coreset selection is often used in cases where the original dataset is too large or complex to be processed efficiently by the available computational resources. By selecting a coreset, practitioners can reduce the computational cost of their analyses and gain valuable insights more efficiently. Data pruning has many interesting applications, notably neural architecture search (NAS), where models trained with a small fraction of the data serve as a proxy to quickly estimate the performance of a given choice of hyper-parameters (Coleman et al., 2020). Another application is continual (or incremental) learning in the context of online learning; To avoid the forgetting problem, one keeps track of the most representative examples of past observations (Aljundi et al., 2019).

Coreset selection is typically performed once during training, and the selected coreset remains fixed until the end of training. This topic has been extensively studied in classical machine learning and statistics (Welling, 2009; Chen, Welling, and Smola, 2012; Feldman, Faulkner, and Krause, 2011; Huggins, Campbell, and Broderick, 2016; Campbell and Broderick, 2019). Recently, many approaches have been proposed to adapt to the challenges of the deep learning context. Examples include removing the redundant examples from the feature space perspective (see Sener and Savarese, 2018), finding the hard examples, defined as the ones for which the model is the least confident (Coleman et al., 2020), or the ones that contribute the most to the error (Toneva et al., 2019). We refer the reader to Section 6 for a more comprehensive literature review. Most of these methods use a score function that ranks examples based on their "importance". Given a desired compression level $r \in (0, 1)$ (the fraction of data kept after pruning), the coreset is created by retaining only the most important examples based on the scores to meet the required compression level. We

---

[*]Equal contribution (Alphabetical order).

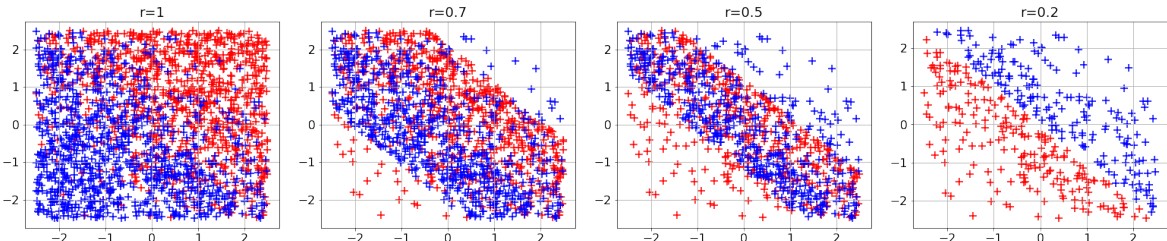

Figure 1: Logistic regression: Data distribution alteration due to pruning for different compression ratios. Here we use `GraNd` as the pruning algorithm. Blue points correspond to $Y_i = 0$, red points correspond to $Y_i = 1$. More details in Section 5.

refer to this type of algorithms as score-based pruning algorithms (`SBPA`). A formal definition is provided in Section 2.

## 1.1 Connection to Neural Scaling Laws

Recently, a stream of empirical works have observed the emergence of power law scaling in different machine learning applications (see e.g. Hestness et al., 2017; Kaplan et al., 2020; Rosenfeld et al., 2020; Hernandez et al., 2021; Zhai et al., 2022; Hoffmann et al., 2022). More precisely, these empirical results show that the performance of the model (e.g. the test error) scales as a power law with either the model size, training dataset size, or compute (FLOPs). In Sorscher et al., 2022, the authors showed that data pruning can improve the power law scaling of the dataset size. The high compression regime (small $r$) is of major interest in this case since it exhibits super-polynomial scaling laws on different tasks. However, as the authors concluded, improving the power law scaling requires high-quality data pruning algorithms, and it is still unclear what properties such algorithms should satisfy. Besides scaling laws, small values of $r$ are of particular interest for tasks such as hyper-parameters selection, where the practitioner wants to select a hyper-parameter from a grid rapidly. In this case, the smaller the value of $r$, the better.

In this work, we argue that score-based data pruning is generally not suited for the high compression regime (starting from $r \leq 30\%$) and, therefore, cannot be used to beat the power law scaling. In this regime, it has been observed (see e.g. Guo, B. Zhao, and Bai, 2022) that most `SBPA` algorithms underperform random pruning (randomly selected subset). To understand why this occurs, we analyze the asymptotic behavior of `SBPA` algorithms and identify some of their properties, particularly in the high compression level regime. To the best of our knowledge, no rigorous explanation for this phenomenon has been reported in the literature. Our work provides the first theoretical explanation for this behavior and offers insights on how to address it in practice.

Intuitively, `SBPA` algorithms induce a distribution shift that affects the training objective. This can, for example, lead to the emergence of new local minima where performance deteriorates significantly. To give a sense of this intuition, we use a toy example in Fig. 1 to illustrate the change in data distribution as the compression level $r$ decreases, where we have used `GraNd` (Paul, Ganguli, and Dziugaite, 2021) to prune the dataset.

We also report the change in the loss landscape in Fig. 2 as the compression level decreases and the resulting scaling laws. The results show that such a pruning algorithm cannot be used to improve the scaling laws since the performance drops significantly in the high compression regime and does not tend to significantly decrease with sample size.

Motivated by these empirical observations, we aim to understand the behaviour of `SBPA` algorithms in the high compression regime. In Section 3, we analyze the impact of pruning of `SBPA` algorithms on the loss function in detail and link this distribution shift to a notion of consistency. We prove several results showing the limitations of `SBPA` algorithms in the high compression regime, which explains some of the empirical results reported in Fig. 2. We also propose calibration protocols, that build on random exploration to address this deterioration in the high compression regime (Fig. 2).

## 1.2 Contributions

Our contributions are as follows:

- We propose a novel formalism to character-ize the asymptotic properties of data prun-ing algorithms in the abundant data regime.

- We introduce Score-Based Pruning Algo-rithms (`SBPA`), a class of algorithms that encompasses a wide range of popular ap-proaches. By employing our formalism, we analyze `SBPA` algorithms and identify a phe-nomenon of distribution shift, which prov-ably impacts generalization error.

- We demonstrate No-Free-Lunch results that characterize when and why score-based pruning algorithms perform worse than ran-dom pruning. Specifically, we prove that `SBPA` are unsuitable for high compression scenarios due to a significant drop in perfor-mance. Consequently, `SBPA` cannot improve scaling laws without appropriate adapta-tion.

- Leveraging our theoretical insights, solu-tions can be designed to address these lim-itations. As an illustration, we introduce a simple calibration protocol to correct the distribution shift by adding noise to the pruning process. Theoretical and empiri-cal results support the effectiveness of this method on toy datasets and show promising results on image classification tasks.[1]

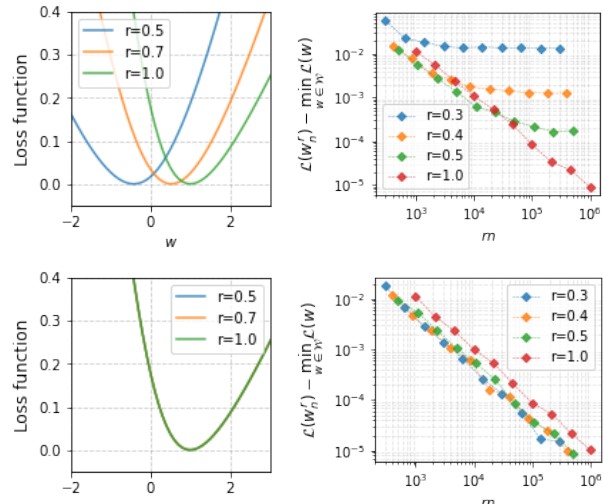

Figure 2: Logistic regression: $r$ is the compression level, $n$ the total number of available data and $w$ the learnable parameter. (**Left**) The loss landscape trans-formation due to pruning. (**Right**) The evolution of the performance gap as the data budget $m := r \times n$ in-creases (average over ten runs). Top figures illustrate the performance of `GraNd`, bottom figures illustrate the performance of `GraNd` calibrated with our exact proto-col: we use 90% of the data budget for the signal, i.e. points selected by `GraNd`, and 10% of the data budget for calibration through random exploration. See Sec-tions 4 and 5 for more details.

## 2 Learning with Data Pruning

### 2.1 Setup

Consider a supervised learning task where the inputs and outputs are respectively in $\mathcal{X} \subset \mathbb{R}^{d_x}$ and $\mathcal{Y} \subset \mathbb{R}^{d_y}$, both assumed to be compact[2]. We denote by $\mathcal{D} = \mathcal{X} \times \mathcal{Y}$ the data space. We assume that there exists $\mu$, an atomless probability distribution on $\mathcal{D}$ from which input/output pairs $Z = (X, Y)$ are drawn independently at random. We call such $\mu$ a *data generating process*. We will assume that $X$ is continuous while $Y$ can be either continuous (regression) or discrete (classification). We are given a family of models

$$\mathcal{M}_\theta = \{y_{out}(\cdot; w) : \mathcal{X} \to \mathcal{Y} \mid w \in \mathcal{W}_\theta\}, \tag{1}$$

parameterised by the *parameter space* $\mathcal{W}_\theta$, a compact subspace of $\mathbb{R}^{d_\theta}$, where $\theta \in \Theta$ is a fixed *hyper-parameter*. For instance, $\mathcal{M}_\theta$ could be a family of neural networks of a given architecture, with weights $w$, and where the architecture is given by $\theta$. We will assume that $y_{out}$ is continuous on $\mathcal{X} \times \mathcal{W}_\theta$[3]. For a given

---

[1]It is important to note that the calibration protocol serves as an example to stimulate further research. We do not claim that this method systematically allows to outperform random pruning nor to beat the neural scaling laws.

[2]We further require that the set $\mathcal{X}$ has no isolated points. This technical assumption is required to avoid dealing with unnecessary complications in the proofs.

[3]This is generally satisfied for a large class of models, including neural networks.

continuous loss function $\ell : \mathcal{Y} \times \mathcal{Y} \to \mathbb{R}$, the aim of the learning procedure is to find a model that minimizes the generalization error, defined by

$$\mathcal{L}(w) \stackrel{def}{=} \mathbf{E}_\mu \, \ell\big(y_{out}(X; w), Y\big). \tag{2}$$

We are given a dataset $\mathcal{D}_n$ composed of $n \geq 1$ input/output pairs $(x_i, y_i)$, *iid* sampled from the data generating process $\mu$. To obtain an approximate minimizer of the generalization error (Eq. (2)), we perform an empirical risk minimization, solving the problem

$$\min_{w \in \mathcal{W}_\theta} \; \mathcal{L}_n(w) \stackrel{def}{=} \frac{1}{n} \sum_{i=1}^{n} \ell\big(y_{out}(x_i; w), y_i\big). \tag{3}$$

The minimization problem (3) is typically solved using a numerical approach, often gradient-based, such as Stochastic Gradient Descent (Robbins and Monro, 1951), Adam (Kingma and Ba, 2017), etc. We refer to this procedure as the *training algorithm*. We assume that the training algorithm is exact, i.e. it will indeed return a minimizing parameter $w_n^* \in \operatorname{argmin}_{w \in \mathcal{W}_\theta} \mathcal{L}_n(w)$. The numerical complexity of the training algorithms grows with the sample size $n$, typically linearly or worse. When $n$ is large, it is appealing to extract a representative *subset* of $\mathcal{D}_n$ and perform the training with this subset, which would reduce the computational cost of training. This process is referred to as data pruning. However, in order to preserve the performance, the subset should retain essential information from the original (full) dataset. This is the primary objective of data pruning algorithms. We begin by formally defining such algorithms.

**Notation.** If $Z$ is a finite set, we denote by $|Z|$ its cardinal number, i.e. the number of elements in $Z$. We denote $\lfloor x \rfloor$ the largest integer smaller than or equal to $x$ for $x \in \mathbb{R}$. For some Euclidean space $\mathcal{E}$, we denote by $d$ the Euclidean distance and for some set $B \subset \mathcal{E}$ and $e \in \mathcal{E}$, we define the distance $d(e, B) = \inf_{b \in B} d(e, b)$. Finally, for two integers $n_1 < n_2$, $[n_1 : n_2]$ refers to the set $\{n_1, n_1 + 1, \ldots, n_2\}$. We denote the set of all finite subsets of $\mathcal{D}$ by $\mathcal{C}$, i.e. $\mathcal{C} = \cup_{n \geq 1}\{\{z_1, z_2, \ldots, z_n\}, z_1 \neq z_2 \neq \ldots \neq z_n \in \mathcal{D}\}$. We call $\mathcal{C}$ the finite power set of $\mathcal{D}$.

**Definition 1 (Data Pruning Algorithm)** *We say that a function $\mathcal{A} : \mathcal{C} \times (0, 1] \to \mathcal{C}$ is a data pruning algorithm if for all $Z \in \mathcal{C}, r \in (0, 1]$, such that $r|Z|$ is an integer [4], we have the following*

- *$\mathcal{A}(Z, r) \subset Z$*

- *$|\mathcal{A}(Z, r)| = r|Z|$*

*where $|.|$ refers to the cardinal number. The number $r$ is called the compression level and refers to the fraction of the data kept after pruning.*

Among the simplest pruning algorithms, we will pay special attention to `Random` pruning, which selects uniformly at random a fraction of the elements of $Z$ to meet some desired compression level $r$.

## 2.2 *Valid* and *Consistent* Pruning Algorithms

Given a pruning algorithm $\mathcal{A}$ and a compression level $r$, a subset of the training set is selected and the model is trained by minimizing the empirical loss on the subset. More precisely, the training algorithm finds a parameter $w_n^{\mathcal{A}, r} \in \operatorname{argmin}_{w \in \mathcal{W}_\theta} \mathcal{L}_n^{\mathcal{A}, r}(w)$ where

$$\mathcal{L}_n^{\mathcal{A}, r}(w) \stackrel{def}{=} \frac{1}{|\mathcal{A}(\mathcal{D}_n, r)|} \sum_{(x, y) \in \mathcal{A}(\mathcal{D}_n, r)} \ell\big(y_{out}(x; w), y\big).$$

---

[4]We make this assumption to simplify the notations. One can take the integer part of $rn$ instead.

This usually requires only a fraction $r$ of the original energy/time[5] cost or better, given the linear complexity of the training algorithm with respect to the data size. In this work, we evaluate the quality of a pruning algorithm by considering the performance gap it induces, i.e. the excess risk of the selected model

$$\text{gap}_n^{\mathcal{A},r} = \mathcal{L}(w_n^{\mathcal{A},r}) - \min_{w \in \mathcal{W}_\theta} \mathcal{L}(w). \tag{4}$$

In particular, we are interested in the abundant data regime: we aim to understand the asymptotic behavior of the performance gap as the sample size $n$ grows to infinity. We define the notion of *valid* pruning algorithms as follows.

**Definition 2 (Valid pruning algorithm)** *For a parameter space $\mathcal{W}_\theta$, a pruning algorithm $\mathcal{A}$ is valid at a compression level $r \in (0, 1]$ if $\lim_{n \to \infty} \text{gap}_n^{\mathcal{A},r} = 0$ almost surely. The algorithm is said to be valid if it is valid at any compression level $r \in (0, 1]$.*

We argue that a valid data pruning algorithm for a given generating process $\mu$ and a family of models $\mathcal{M}_\theta$ should see its performance gap converge to zero almost surely. Otherwise, it would mean that with positive probability, the pruning algorithm induces a deterioration of the out-of-sample performance that does not vanish even when an arbitrarily large amount of data is available. This deterioration would not exist without pruning or if random pruning was used instead (Corollary 1). This means that with positive probability, a non-valid pruning algorithm will underperform random pruning in the abundant data regime. In the next result, we show that a sufficient and necessary condition for a pruning algorithm to be valid at compression level $r$ is that $w_n^{\mathcal{A},r}$ should approach the set of minimizers of the original generalization loss function as $n$ increases.

**Proposition 1 (Characterization of valid pruning algorithms)** *A pruning algorithm $\mathcal{A}$ is valid at a compression level $r \in (0, 1]$ if and only if*

$$d\left(w_n^{\mathcal{A},r}, \mathcal{W}_\theta^*(\mu)\right) \to 0 \ a.s.$$

*where $\mathcal{W}_\theta^*(\mu) = \text{argmin}_{w \in \mathcal{W}_\theta} \mathcal{L}(w) \subset \mathcal{W}_\theta$ and $d\left(w_n^{\mathcal{A},r}, \mathcal{W}_\theta^*(\mu)\right)$ denotes the euclidean distance from the point $w_n^{\mathcal{A},r}$ to the set $\mathcal{W}_\theta^*(\mu)$.*

With this characterization in mind, the following proposition provides a key tool to analyze the performance of pruning algorithms. Under some conditions, it allows us to describe the asymptotic performance of any pruning algorithm via some properties of a probability measure.

**Proposition 2** *Let $\mathcal{A}$ be a pruning algorithm and $r \in (0, 1]$ a compression level. Assume that there exists a probability measure $\nu_r$ on $\mathcal{D}$ such that*

$$\forall w \in \mathcal{W}_\theta, \ \mathcal{L}_n^{\mathcal{A},r}(w) \to \mathbb{E}_{\nu_r} \ell(y_{out}(X; w), Y) \ a.s. \tag{5}$$

*Then, denoting $\mathcal{W}_\theta^*(\nu_r) = \text{argmin}_{w \in \mathcal{W}_\theta} \mathbb{E}_{\nu_r} \ell(y_{out}(X; w), Y) \subset \mathcal{W}_\theta$, we have that*

$$d\left(w_n^{\mathcal{A},r}, \mathcal{W}_\theta^*(\nu_r)\right) \to 0 \ a.s.$$

Condition Eq. (5) assumes the existence of a limiting probability measure $\nu_r$ that represents the distribution of the pruned dataset in the limit of infinite sample size. In Section 3, for a large family of pruning algorithms called score-based pruning algorithms (a formal definition will be introduced later), we will demonstrate the existence of such limiting probability measure and derive its exact expression.

Let us now derive two important corollaries; the first gives a sufficient condition for an algorithm to be valid, and the second a necessary condition. From Proposition 1 and Proposition 2, we can deduce that a sufficient condition for an algorithm to be valid is that $\nu_r = \mu$ satisfies equation (5). We say that such a pruning algorithm is *consistent*.

---

[5]Here the original cost refers to the training cost of the model with the full dataset.

**Definition 3 (Consistent Pruning Algorithms)** *We say that a pruning algorithm $\mathcal{A}$ is consistent at compression level $r \in (0, 1]$ if and only if it satisfies*

$$\forall w \in \mathcal{W}_\theta, \ \mathcal{L}_n^{\mathcal{A},r}(w) \to \mathbb{E}_\mu[\ell(y_{out}(x,w), y)] = \mathcal{L}(w) \ a.s. \tag{6}$$

*We say that $\mathcal{A}$ is consistent if it is consistent at any compression level $r \in (0, 1]$.*

**Corollary 1** *A consistent pruning algorithm $\mathcal{A}$ at a compression level $r \in (0, 1]$ is also valid at compression level $r$.*

A simple application of the law of large numbers implies that `Random` pruning is consistent and hence valid for any generating process and learning task satisfying our general assumptions.

We bring to the reader's attention that consistency is itself a property of practical interest. Indeed, it not only ensures that the generalization gap of the learned model vanishes, but it also allows the practitioner to accurately estimate the generalization error of their trained model from the selected subset. For instance, consider the case where the practitioner is interested in $K$ hyper-parameter values $\theta_1, ..., \theta_K$; these can be different neural network architectures (depth, width, etc.). Using a pruning algorithm $\mathcal{A}$, they obtain a trained model $w_n^{\mathcal{A},r}(\theta_k)$ for each hyper-parameter $\theta_k$, with corresponding estimated generalization error $\mathcal{L}_n^{\mathcal{A},r}\left(w_n^{\mathcal{A},r}(\theta_k)\right)$. Hence, the consistency property would allow the practitioner to select the best hyper-parameter value based on the empirical loss computed with the set of retained points (or a random subset of which used for validation). From Proposition 1 and Proposition 2, we can also deduce a necessary condition for an algorithm satisfying (5) to be valid:

**Corollary 2** *Let $\mathcal{A}$ be any pruning algorithm and $r \in (0, 1]$, and assume that (5) holds for a given probability measure $\nu_r$ on $\mathcal{D}$. If $\mathcal{A}$ is valid, then $\mathcal{W}_\theta^*(\nu_r) \cap \mathcal{W}_\theta^*(\mu) \neq \emptyset$; or, equivalently,*

$$\min_{w \in \mathcal{W}_\theta^*(\nu_r)} \mathcal{L}(w) = \min_{w \in \mathcal{W}} \mathcal{L}(w).$$

Corollary 2 will be a key ingredient in the proofs on the non-validity of a given pruning algorithm. Specifically, for all the non-validity results stated in this paper, we prove that $\mathcal{W}_\theta^*(\nu_r) \cap \mathcal{W}_\theta^*(\mu) = \emptyset$. In other words, none of the minimizers of the original problem is a minimizer of the pruned one, and vice-versa.

## 3 Score-Based Pruning Algorithms and their Limitations

### 3.1 Score-based Pruning Algorithms

A standard approach to define a pruning algorithm is to assign to each sample $z_i = (x_i, y_i)$ a score $g_i = g(z_i)$ according to some *score function $g$*, where $g$ is a mapping from $\mathcal{D}$ to $\mathbb{R}$. $g$ is also called the pruning criterion. The score function $g$ captures the practitioner's prior knowledge of the relative importance of each sample. This function can be defined using a teacher model that has already been trained, for example. In this work, we use the convention that the lower the score, the more relevant the example. One could of course adopt the opposite convention by considering $-g$ instead of $g$ in the following. We now formally define this category of pruning algorithms, which we call score-based pruning algorithms.

**Definition 4 (Score-based Pruning Algorithm (`SBPA`))** *Let $\mathcal{A}$ be a data pruning algorithm. We say that $\mathcal{A}$ is a score-based pruning algorithm (`SBPA`) if there exists a function $g : \mathcal{D} \to \mathbb{R}$ such that for all $Z \in \mathcal{C}$, $r \in (0, 1)$, we have that $\mathcal{A}(Z, r) = \{z \in Z, \text{s.t. } g(z) \leq g^{r|Z|}\}$, where $g^{r|Z|}$ is $(r|Z|)^{th}$ order statistic of the sequence $(g(z))_{z \in Z}$ (first order statistic being the smallest value). The function $g$ is called the score function.*

A significant number of existing data pruning algorithms are score-based (for example **Coleman2020Uncertainty**; Paul, Ganguli, and Dziugaite, 2021; Ducoffe and Precioso, 2018; Sorscher et al., 2022), among which the recent approaches for modern machine learning. One of the key benefits of

these methods is that the scores are computed independently; these methods are hence parallelizable, and their complexity scales linearly with the data size (up to log terms). These methods are tailored for the abundant data regime, which explains their recent gain in popularity.

Naturally, the result of such a procedure highly depends on the choice of the score function $g$, and different choices of $g$ might yield completely different subsets. The choice of the score function in Definition 4 is not restricted, and there are many scenarios in which the selection of the score function $g$ may be problematic. For example, if $g$ has discontinuity points, this can lead to instability in the pruning procedure, as close data points may have very different scores. Another problematic scenario is when $g$ assigns the same score to a large number of data points. To avoid such unnecessary complications, we define *adapted* pruning criteria as follows:

**Definition 5 (Adapted score function)** *Let $g$ be a score function corresponding to some pruning algorithm $\mathcal{A}$. We say that $g$ is an adapted score function if $g$ is continuous and for any $c \in g(\mathcal{D}) := \{g(z), z \in \mathcal{D}\}$, we have $\lambda(g^{-1}(\{c\})) = 0$, where $\lambda$ is the Lebesgue measure on $\mathcal{D}$.*

In the rest of the section, we will examine the properties of `SBPA` algorithms with an adapted score function.

### 3.2 Asymptotic Behavior of `SBPA`

Asymptotically, `SBPA` algorithms have a simple behavior that mimics rejection algorithms. We describe this in the following result.

**Proposition 3 (Asymptotic behavior of `SBPA`)** *Let $\mathcal{A}$ be a `SBPA` algorithm and let $g$ be its corresponding adapted score function. Consider a compression level $r \in (0,1)$. Denote by $q^r$ the $r^{th}$ quantile of the random variable $g(Z)$ where $Z \sim \mu$. Denote $A_r = \{z \in \mathcal{D} \mid g(z) \leq q^r\}$. Almost surely, the empirical measure of the retained data samples converges weakly to $\nu_r = \frac{1}{r}\mu_{|A_r}$, where $\mu_{|A_r}$ is the restriction of $\mu$ to the set $A_r$. In particular, we have that*

$$\forall w \in \mathcal{W}_\theta, \ \mathcal{L}_n^{\mathcal{A},r}(w) \to \mathbb{E}_{\nu_r}\ell(y_{out}(X;w),Y) \ a.s.$$

The result of Proposition 3 implies that in the abundant data regime, a `SBPA` algorithm $\mathcal{A}$ acts similarly to a deterministic rejection algorithm, where the samples are retained if they fall in $A_r$, and removed otherwise. The first consequence is that a `SBPA` algorithm $\mathcal{A}$ is consistent at compression level $r$ if and only if

$$\forall w \in \mathcal{W}_\theta, \ \mathbb{E}_{\frac{1}{r}\mu_{|A_r}}\ell(y_{out}(X;w),Y) = \mathbb{E}_\mu\ell(y_{out}(X;w),Y), \tag{7}$$

The second consequence is that `SBPA` algorithms ignore entire regions of the data space, even when we have access to unlimited data, i.e. $n \to \infty$. Moreover, the ignored region can be made arbitrarily large for small enough compression levels. Therefore, we expect that the generalization performance will be affected and that the drop in performance will be amplified with smaller compression levels, regardless of the sample size $n$. This hypothesis is empirically validated (see Guo, B. Zhao, and Bai, 2022 and Section 5).

In the rest of the section, we investigate the fundamental limitations of `SBPA` in terms of consistency and validity; we will show that under mild assumptions, for any `SBPA` algorithm with an adapted score function, there exist compression levels $r$ for which the algorithm is neither consistent nor valid. Due to the prevalence of classification problems in modern machine learning, we focus on the binary classification setting and give specialized results in Section 3.3. In Section 3.4, we provide a different type of non-validity results for more general problems.

### 3.3 Binary Classification Problems

In this section, we focus our attention on binary classification problems. The predictions and labels are in $\mathcal{Y} = [0,1]$. Denote $\mathcal{P}_B$ the set of probability distributions on $\mathcal{X} \times \{0,1\}$, such that the marginal distribution on the input space $\mathcal{X}$ is continuous (absolutely continuous with respect to the Lebesgue measure on $\mathcal{X}$) and for which

$$p_\pi : x \mapsto \mathbb{P}_\pi(Y = 1|X = x)$$

is upper semi-continuous for any $\pi \in \mathcal{P}_B$. We further assume that:

(i) the loss is non-negative and that $\ell(y, y') = 0$ if and only if $y = y'$.

(ii) For $q \in [0, 1]$, $y \mapsto q\ell(y, 1) + (1 - q)\ell(y, 0)$ has a unique minimizer, denoted $y_q^* \in [0, 1]$, that is increasing with $q$.

These two assumptions are generally satisfied in practice for the usual loss functions, such as the $\ell_1$, $\ell_2$, Exponential or Cross-Entropy losses, with the notable exception of the Hinge loss for which (ii) does not hold.

Under mild conditions that are generally satisfied in practice, we show that no `SBPA` algorithm is consistent. We first define a notion of universal approximation.

**Definition 6 (Universal approximation)** *A family of continuous functions $\Psi$ has the universal approximation property if for any continuous function $f : \mathcal{X} \to \mathcal{Y}$ and $\epsilon > 0$, there exists $\psi \in \Psi$ such that*

$$\max_{x \in \mathcal{X}} |f(x) - \psi(x)| \leq \epsilon$$

The next proposition shows that if the set of all models considered $\cup_{\theta \in \Theta} \mathcal{M}_\theta$ has the universal approximation property, then no `SBPA` algorithm is consistent.

**Theorem 1** *Consider any generating process for binary classification $\mu \in \mathcal{P}_B$. Let $\mathcal{A}$ be any `SBPA` algorithm with an adapted score function. If $\cup_\theta \mathcal{M}_\theta$ has the universal approximation property and the loss satisfies assumption (i), then there exist hyper-parameters $\theta \in \Theta$ for which the algorithm is not consistent.*

Even though consistency is an important property, a pruning algorithm can still be valid without being consistent. In this classification setting, we can further show that `SBPA` algorithms also have strong limitations in terms of validity.

**Theorem 2** *Consider any generating process for binary classification $\mu \in \mathcal{P}_B$. Let $\mathcal{A}$ be a `SBPA` with an adapted score function $g$ that depends on the labels[6]. If $\cup_\theta \mathcal{M}_\theta$ has the universal approximation property and the loss satisfies assumptions (i) and (ii), then there exist hyper-parameters $\theta_1, \theta_2 \in \Theta$ and $r_0 \in (0, 1)$ such that the algorithm is not valid for $r \leq r_0$ for any hyper-parameter $\theta$ such that $\mathcal{W}_{\theta_1} \cup \mathcal{W}_{\theta_2} \subset \mathcal{W}_\theta$.*

This theorem sheds light on a strong limitation of `SBPA` algorithms for which the score function depends on the labels: it states that any solution of the pruned program will induce a generalization error strictly larger than with random pruning in the abundant data regime. The proof builds on Corollary 2; we show that for such hyper-parameters $\theta$, the minimizers of the pruned problem and the ones of the original (full data) problem do not intersect, i.e.

$$\mathcal{W}_\theta^*(\nu_r) \cap \mathcal{W}_\theta^*(\mu) = \emptyset.$$

`SBPA` algorithms usually depend on the labels (**Coleman2020Uncertainty**; Paul, Ganguli, and Dziugaite, 2021; Ducoffe and Precioso, 2018) and Theorem 2 applies. In Sorscher et al., 2022, the authors also propose to use a `SBPA` that does not depend on the labels. For such algorithms, the acceptance region $A_r$ is characterized by a corresponding input acceptance region $\mathcal{X}_r$. `SBPA` independent of the labels have a key benefit; the conditional distribution of the output is not altered given that the input is in $\mathcal{X}_r$. Contrary to the algorithms depending on the labels, the performance will not necessarily be degraded for any generating distribution given that the family of models is rich enough. It remains that the pruned data give no information outside of $\mathcal{X}_r$, and $y_{out}$ can take any value in $\mathcal{X} \setminus \mathcal{X}_r$ without impacting the pruned loss. Hence, these algorithms can create new local/global minima with poor generalization performance. Besides, the non-consistency results of this section and the No-Free-Lunch result presented in Section 3.4 do apply for `SBPA` independent of the labels. For these reasons, we believe that calibration methods (see Section 4) should also be employed for `SBPA` independent of the labels, especially with small compression ratios.

---

[6]The score function $g$ depends on the labels if there exists an input $x$ in the support of the distribution of the input $X$ and for which $g(x, 0) \neq g(x, 1)$ and $\mathbb{P}(Y = 1 \mid X = x) \in (0, 1)$ (both labels can happen at input $x$)

**Applications: Neural Networks**

To exemplify the utility of Theorem 1 and Theorem 2, we leverage the existing literature on the universal approximation properties of neural networks to derive the important corollaries stated below

**Definition 7** *For an activation function $\sigma$, a real number $R > 0$, and integers $H, K \geq 1$, we denote by $FFNN_{H,K}^{\sigma}(R)$ the set of fully-connected feed-forward neural networks with $H$ hidden layers, each with $K$ neurons with all weights and biases in $[-R, R]$.*

**Corollary 3 (Wide neural networks)** *Let $\sigma$ be any continuous non-polynomial function that is continuously differentiable at (at least) one point, with a nonzero derivative at that point. Consider any generating process $\mu \in \mathcal{P}_B$. For any SBPA with adapted score function and $H \geq 1$, there exists a radius $R_0$ and a width $K_0$ such that the algorithm is not consistent on $FFNN_{H,K}^{\sigma}(R)$ for any $K \geq K_0$ and $R \geq R_0$. Besides, if the score function depends on the labels, then it is also not valid on $FFNN_{H,K}^{\sigma}(R)$ for any $K \geq K_0'$ and $R \geq R_0'$.*

**Corollary 4 (Deep neural networks)** *Consider a width $K \geq d_x + 2$. Let $\sigma$ be any continuous non-polynomial function that is continuously differentiable at (at least) one point, with a nonzero derivative at that point. Consider any generating process $\mu \in \mathcal{P}_B$. For any SBPA with an adapted score function, there exists a radius $R_0$ and a number of hidden layers $H_0$ such that the algorithm is not consistent on $FFNN_{H,K}^{\sigma}(R)$ for any $H \geq H_0$ and $R \geq R_0$. Besides, if the score function depends on the labels, then it is also not valid on $FFNN_{H,K}^{\sigma}(R)$ for any $H \geq H_0'$ and $R \geq R_0'$*

A similar result for convolutional architectures is provided in Appendix C. To summarize, these corollaries show that for large enough neural network architectures, any SBPA is non-consistent. Besides, for large enough neural network architectures, any SBPA that depends on the label is non-valid, and hence a performance gap should be expected even in the abundant data regime.

### 3.4 General Problems

In the previous section, we leveraged the universal approximation property and proved non-validity and non-consistency results that hold for any data-generating process. In this section, we show a different No-free-Lunch result in the general setting presented in Section 2. This result does not require the universal approximation property. More precisely, we show that under mild assumptions, given any SBPA algorithm, we can always find a data distribution $\mu$ such that the algorithm is not valid (Definition 2). Since random pruning is valid for any generating process, this means that there exist data distributions for which the SBPA algorithm provably underperforms random pruning in the abundant data regime.

For $K \in \mathbb{N}^*$, let $\mathcal{P}_C^K$ denote the set of generating processes for $K$-classes classification problems, for which the input $X$ is a continuous random variable[7], and the output $Y$ can take one of $K$ values in $\mathcal{Y}$ (the same set of values for all $\pi \in \mathcal{P}_C^K$). Similarly, denote $\mathcal{P}_R$, the set of generating processes for regression problems for which both the input and output distributions are continuous. Let $\mathcal{P}$ be any set of generating processes introduced previously for regression or classification (either $\mathcal{P} = \mathcal{P}_C^K$ for some $K$, or $\mathcal{P} = \mathcal{P}_R$). In the next theorem, we show that under minimal conditions, there exists a data generating process for which the algorithms is not valid.

**Theorem 3** *Let $\mathcal{A}$ be a SBPA with an adapted score function. For any hyper-parameter $\theta \in \Theta$, if there exist $(x_1, y_1), (x_2, y_2) \in \mathcal{D}$ such that*

$$\text{argmin}_{w \in \mathcal{W}_\theta} \ell(y_{out}(x_1; w), y_1) \cap \text{argmin}_{w \in \mathcal{W}_\theta} \ell(y_{out}(x_2; w), y_2) = \emptyset, \tag{H1}$$

*then there exists $r_0 \in (0, 1)$ and a generating process $\mu \in \mathcal{P}$ for which the algorithm is not valid for $r \leq r_0$.*

The rigorous proof of Theorem 3 requires careful manipulations of different quantities, but the intuition is rather simple. Fig. 3 illustrates the main idea of the proof. We construct a distribution $\mu$ with the majority

---

[7]In the sense that the marginal of the input is dominated by the Lebesgue measure

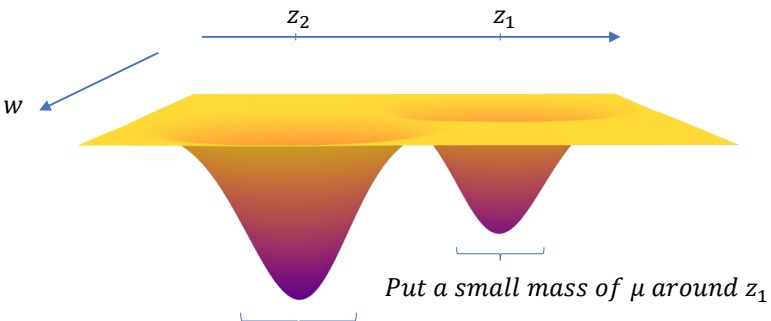

Figure 3: Graphical sketch of the proof of Theorem 3. The surface represents the loss function $f(z, w) = \ell(y_{out}(x), y)$ in 2D, where $z = (x, y)$.

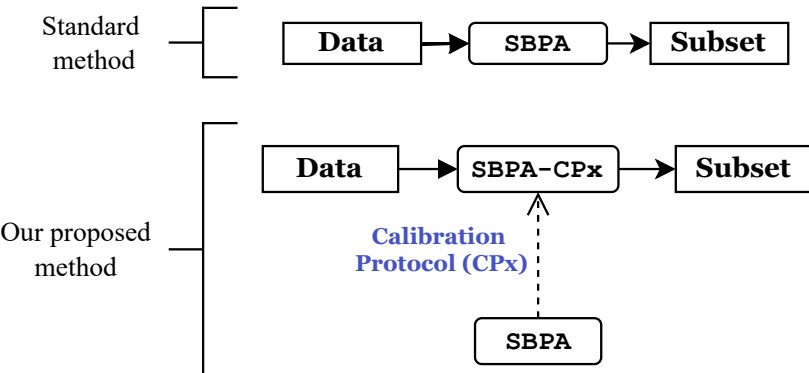

Figure 4: An illustration of how the calibration protocols modify SBPA algorithms.

of the probability mass concentrated around a point where the value of $g$ is not minimal. Consequently, for sufficiently small $r$, the distribution of the retained samples will significantly differ from the original distribution. This shift in data distributions causes the algorithm to be non-valid. We see in the next section how we can solve this issue via randomization. Finally, notice that Eq. (H1) is generally satisfied in practice since usually for two different examples $(x_1, y_1)$ and $(x_2, y_2)$ in the datasets, the global minimizers of $\ell(y_{out}(x_1; w), y_1)$ and $\ell(y_{out}(x_2; w), y_2)$ are different.

## 4   Solving Non-Consistency via Randomization

We have seen in Section 3 that SBPA algorithms inherently transform the data distribution by asymptotically rejecting all samples in $\mathcal{D} \setminus A_r$. These algorithms are prone to inconsistency; the transformation of the data distribution translates to a distortion of the loss landscape, potentially leading to a deterioration of the generalization error. This effect is exacerbated for smaller compression ratios $r$ as the acceptance region becomes arbitrarily small and concentrated.

With this in mind, one can design practical solutions to mitigate the problem. For illustration, we propose to resort to a *Calibration Protocol* to retain information from the previously discarded region $\mathcal{D} \setminus A_r$. The calibration protocols can be thought of as wrapper modules that can be applied on top of any `SBPA` algorithm to solve the consistency issue through randomization (see Fig. 4 for a graphical illustration). Specifically, we split the data budget $rn$ into two parts: the first part, allocated for the *signal*, leverages the knowledge from the `SBPA` and its score function $g$. The second part, allocated for *exploration*, accounts for the discarded region and consists of a subset of the rejected points, selected uniformly at random. In other words, we write $r = r_{signal} + r_{exploration}$. With standard `SBPA` procedures, $r_{exploration} = 0$. We define $\alpha = \frac{r_{signal}}{r}$ the proportion of signal in the overall budget. Accordingly, the set of retained points can be expressed as

$$\bar{\mathcal{A}}(\mathcal{D}_n, r, \alpha) = \bar{\mathcal{A}}_s(\mathcal{D}_n, r, \alpha) \cup \bar{\mathcal{A}}_e(\mathcal{D}_n, r, \alpha),$$

where $\bar{\mathcal{A}}$ denotes the calibrated version of $\mathcal{A}$, and the indices 's' and 'e' refer to signal and exploration respectively. In this work, we consider the simplest approach. The "signal subset" is composed of the $\alpha rn$ points with the highest importance according to $g$, i.e. $\bar{\mathcal{A}}_s(\mathcal{D}_n, r, \alpha) = \mathcal{A}(\mathcal{D}_n, r\alpha)$. The "exploration subset", $\bar{\mathcal{A}}_e(\mathcal{D}_n, r, \alpha)$ is composed on average of $(1-\alpha)rn$ points selected uniformly at random from the remaining samples $\mathcal{D}_n \setminus \mathcal{A}(\mathcal{D}_n, r\alpha)$, each sample being retained with probability $p_e = \frac{(1-\alpha)r}{1-\alpha r}$, independently. The calibrated loss is then defined as a weighted sum of the contributions of the signal and exploration budgets,

$$\mathcal{L}_n^{\bar{\mathcal{A}}, r, \alpha}(w) = \frac{1}{n} \left( \gamma_s \sum_{z \in \bar{\mathcal{A}}_s(\mathcal{D}_n, r, \alpha)} f(z; w) + \gamma_e \sum_{z \in \bar{\mathcal{A}}_e(\mathcal{D}_n, r, \alpha)} f(z; w) \right) \tag{8}$$

where $f(z; w) = \ell(y_{out}(x), y)$ for $z = (x, y) \in \mathcal{D}$ and $w \in \mathcal{W}_\theta$. The weights $\gamma_s$ and $\gamma_e$ are chosen so that the calibrated procedure is consistent; they are inversely proportional to the probability of acceptance within each region:

$$\begin{aligned} \gamma_s &= 1 \\ \gamma_e &= \frac{1 - \alpha r}{(1 - \alpha)r} \end{aligned}$$

Proposition 4 hereafter states that any `SBPA` calibrated with this procedure is made consistent as long as a non-zero budget is allocated to exploration. For this reason, we refer to this method as the Exact Calibration protocol (EC). The proof builds on an adapted version of the law of large numbers for sequences of dependent variables which we prove in the Appendix (Theorem 7).

**Proposition 4 (Consistency of Exact Calibration+`SBPA`)** *Let $\mathcal{A}$ be a `SBPA` algorithm. Using the Exact Calibration protocol with signal proportion $\alpha$, the calibrated algorithm $\bar{\mathcal{A}}$ is consistent if $1 - \alpha > 0$, i.e. the exploration budget is not null. Besides, under the same assumption $1 - \alpha > 0$, the calibrated loss is an unbiased estimator of the generalization loss at any finite sample size $n > 0$,*

$$\forall w \in \mathcal{W}_\theta, \ \forall r \in (0, 1), \ \mathbb{E}\mathcal{L}_n^{\bar{\mathcal{A}}, r, \alpha}(w) = \mathcal{L}(w).$$

The proposed EC protocol offers a simple yet effective approach to address the challenges of non-consistency and non-validity. It can be seamlessly applied in conjunction with any `SBPA`. The core concept revolves around the implementation of soft-pruning: any data point is assigned a non-zero selection probability. Samples with lower scores are given a higher acceptance rate. The contribution of each accepted data point to the loss is then weighted accordingly. The EC protocol embodies one specific implementation of soft-pruning, offering the advantage of a single interpretable tuning parameter, the signal proportion $\alpha \in [0, 1]$. By setting $\alpha$ to 1 or 0, one can recover the `SBPA` and `Random` pruning as extreme cases.

Proposition 4 states that any `SBPA` calibrated with EC is made consistent and valid as long as some budget is allocated to exploration. This is empirically validated in Section 5 where we show promising results on a Toy example (Logistic regression) and other image tasks. However, the exact calibration protocol does not systematically allow to outperform random pruning.

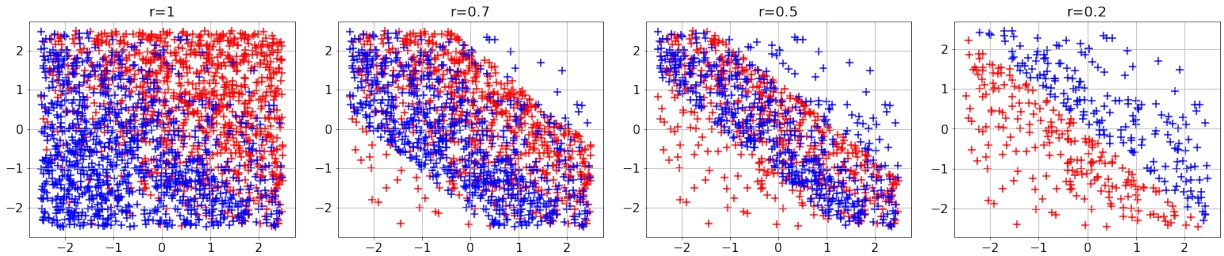

Figure 5: Data distribution alteration due to pruning in the logistic regression setting. Here we use `GraNd` as the pruning algorithm. Blue points correspond to $Y_i = 0$, red points correspond to $Y_i = 1$.

Besides, it is worth noting that different implementations of the same general recipe can be considered. It is reasonable to expect that more tailored protocols can be designed to suit specific pruning algorithms and problems. Nevertheless, addressing these questions falls outside the scope of the present work which focus is to provide a framework to analyse data pruning algorithms, as well as to identify and understand their fundamental limitations. We propose the EC protocol to illustrate how this understanding allows to design simple yet efficient solutions to address these limitations.

## 5 Experiments

### 5.1 Logistic Regression

We illustrate the main results of this work on a logistic regression task. We consider the following data-generating process

$$X_i \sim \mathcal{U}\big([-2.5, 2.5]^{d_x}\big)$$
$$Y_i \mid X_i \sim \mathcal{B}\left(\frac{1}{1 + e^{-w_0^T X_i}}\right),$$

where $w_0 = (1, ..., 1) \in \mathbb{R}^{d_x}$, $\mathcal{U}$ and $\mathcal{B}$ are respectively the uniform and Bernoulli distributions. The class of models is given by

$$\mathcal{M} = \left\{ y_{out}(\cdot; w) : x \mapsto \frac{1}{1 + e^{-w^T X_i}} \mid w \in \mathcal{W} \right\},$$

where $\mathcal{W} = [-10, 10]^{d_x}$. We train the models using stochastic gradient descent with the cross entropy loss. For performance analysis, we take $d_x = 20$ and $n = 10^6$. For the sake of visualization, we take $d_x = 1$ when we plot the loss landscapes (so that the parameter $w$ is univariate) and $d_x = 2$ when we plot the data distributions.

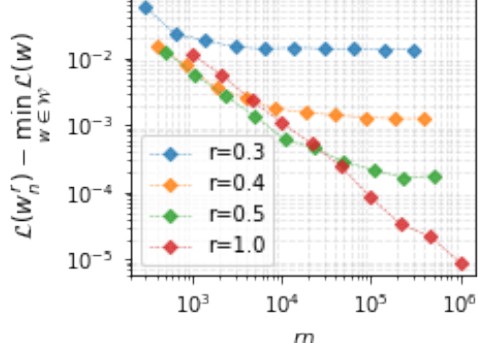

Figure 6: Evolution of the performance gap as the data budget $m = rn$ increases (average over 10 runs).

We use `GraNd` (Paul, Ganguli, and Dziugaite, 2021) as a pruning algorithm in a teacher-student setting. For simplicity, we use the optimal model to compute the scores, i.e.

$$g(X_i, Y_i) = -\|\nabla_w \ell(y_{out}(X_i, w_0), Y_i)\|^2,$$

which is proportional to $-(y_{out}(X_i; w_0) - Y_i)^2$. Notice that in this setting, `GraNd` and `EL2N` (Paul, Ganguli, and Dziugaite, 2021) are equivalent[8]. We bring to the reader's attention that $r = 1$ corresponds to `Random` pruning in our plots. Indeed, we compare models as a function of the data budget $m = rn$. But notice that

---

[8]This is different from the original version of `GraNd` , here, we have access to the true generating process, which is not the case in practice.

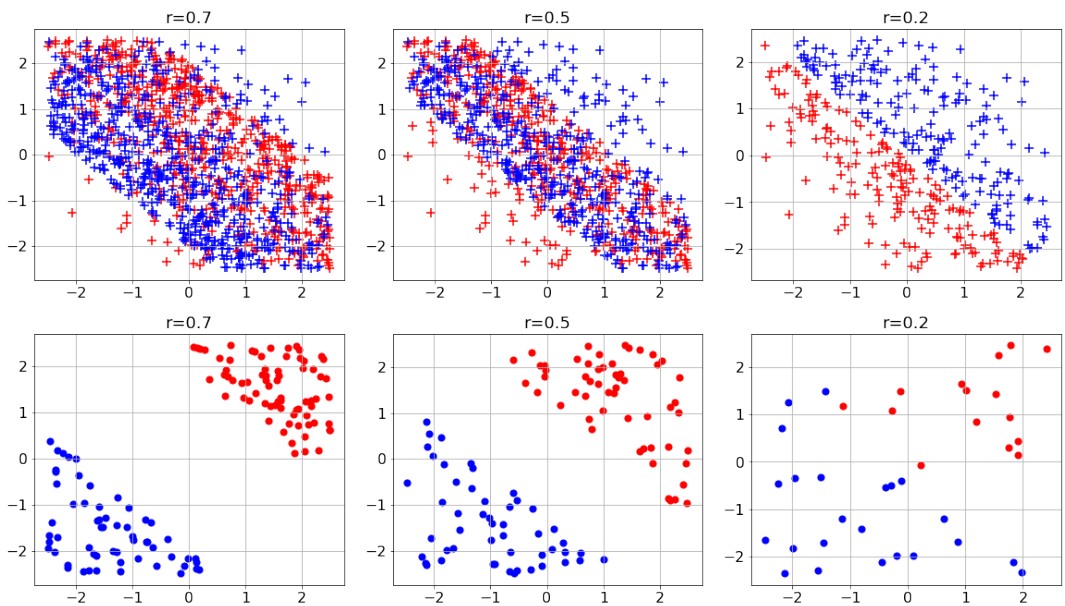

Figure 8: Pruned data distribution for `GraNd` calibrated with exact protocol with $\alpha = 90\%$. The top figures represent the 'signal' points. The bottom figures represent the 'exploration' points. Blue markers correspond to $Y_i = 0$, and red markers correspond to $Y_i = 1$.

in the case of `Random`, for a given $m$, the values of $r$ and $n$ do not affect the distribution of the accepted datapoints, and this distribution is always the same as the original data distribution, i.e. when $r = 1$.

**Distribution shift and performance degradation:** In Section 3, we have seen that the pruning algorithm induces a shift in the data distribution (Fig. 5). This alteration is most pronounced when $r$ is small; For $r = 20\%$, the bottom-left part of the space is populated by $Y = 1$ and the top-right by $Y = 0$. Notice that it was the opposite in the original dataset ($r = 1$). This translates into a distortion of the loss landscape and the optimal parameters $w_n^{\mathcal{A},r}$ of the pruned empirical loss becomes different from $w_0 = 1$. Hence, even when a large amount of data is available, the performance gap does not vanish (Fig. 6).

**Calibration with the exact protocol:** To solve the distribution shift, we resort to the exact protocol with $\alpha = 90\%$. In other words, 10% of the budget is allocated to exploration. The signal points (top images in Fig. 8) are balanced with the exploration points (bottom images in Fig. 8). Even though there are nine times fewer of them, the importance weights allow to correct the distribution shift, as depicted in Fig. 2 (Introduction): the empirical losses overlap for all values of $r$, even for small values for which the predominant labels are swapped (for example $r = 20\%$). Hence, the performance gap vanishes when enough data is available at any compression ratio (Fig. 7).

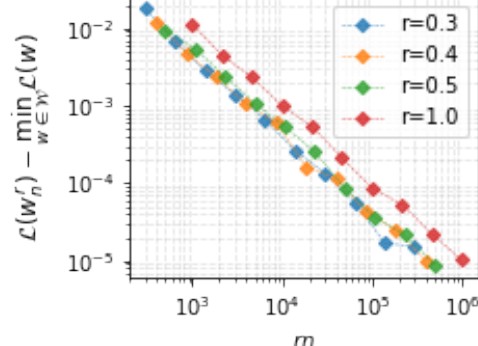

Figure 7: Evolution of the performance gap with calibrated `GraNd` as the data budget $m = rn$ increases (average over 10 runs).

**Impact of the quality of the pruning algorithm:** The calibration protocols allow the performance gap to eventually vanish if enough data is provided. However, from a practical point of view, a natural further requirement is that the pruning method should be better than `Random`, in the sense that for a given finite budget $rn$, the error with the pruning algorithm should be lower than the one of `Random`. We argue that this mostly decided by the quality of the original `SBPA` and its score function. Let us take a closer look at what happens in the

logistic regression case. For a given $X_i$, denote $\tilde{Y}_i$ the most probable label for the input, i.e. $\widetilde{Y}_i = 1$ if $y_{out}(X_i, w_0) > 1/2$, and $\widetilde{Y}_i = 0$ otherwise. As explained, in this setting, GraNd is equivalent to using the score function $g(Z_i) = -|Y_i - y_{out}(X_i; w_0)|$. For a given value of $r$, consider $q^r$ the $r^{th}$ quantile of $g(Z)$. Notice that $g(Z) \leq q^r$ if and only if

$$\underbrace{\left( \left| y_{out}(X_i; w_0) - \frac{1}{2} \right| \leq q^r + \frac{1}{2} \right)}_{\text{Condition 1}} \text{ or } \underbrace{\left( \left| y_{out}(X_i; w_0) - \frac{1}{2} \right| > \left| q^r + \frac{1}{2} \right| \text{ and } Y_i \neq \widetilde{Y}_i \right)}_{\text{Condition 2}}$$

Therefore, the signal acceptance region is the union of two disjoint sets. The first set is composed of all samples that are close to the decision boundary, i.e. samples for which the true conditional probability $y_{out}(X_i; w_0)$ is close to $1/2$. The second set is composed of samples that are further away from the decision boundary, but the realized labels need to be the least probable ones ($Y_i \neq \widetilde{Y}_i$). These two subsets are visible in Figs. 5 and 8 for $r = 70\%$ and even more for $r = 50\%$. The signal points can be divided into two sets:

1. the set of points close to the boundary line $y = -x$, where the colors match the original configurations (mostly blue points under the line, red points over the line)

2. the set of points far away from the boundary line, for which the colors are swapped (only red under the line, blue over the line).

Hence, the signal subset corresponding to Condition 1 gives valuable insights; it provides finer-grained visibility in the critical region. However, the second subset is unproductive, as it only retains points that are not representative of their region. Calibration allows mitigating the effect of the second subset while preserving the benefits of the first subset; in Fig. 7, we can see that the calibrated GraNd outperforms random pruning (which corresponds to the $r = 1$ curve), requiring on average two to three times fewer data to achieve the same generalization error. However, as $r$ becomes lower, $q^r$ will eventually fall under $-1/2$, and the first subset becomes empty (for example, $r = 0.2$ in Fig. 8). Therefore, when $r$ becomes small, GraNd does not bring valuable information anymore (for this particular setting). In Fig. 9, we compare GraNd and Calibrated GraNd (with the exact protocol) to Random with $r = 10\%$. We can see that thanks to the calibration protocol, the performance gap

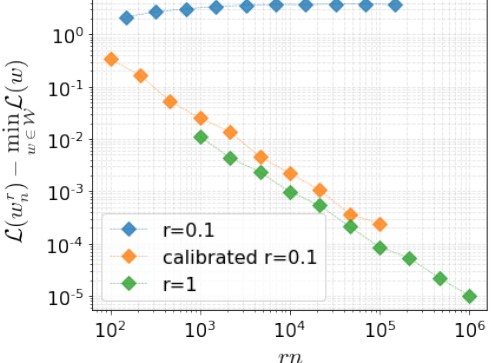

Figure 9: Evolution of the performance gap for a small value $r = 0.1$ for GraNd and its calibrated version with $\alpha = 90\%$.

will indeed vanish if enough data is available. However, Random pruning outperforms both versions of GraNd at this compression level. This underlines the fact that for high compression levels, (problem-specific) high-quality pruning algorithms and score functions are required. Given the difficulty of the task, we believe that in the high compression regime ($r \leq 10\%$ here), one should allocate a larger budget to random exploration (take smaller values of $\alpha$).

## 5.2 Scaling Laws with Neural Networks

The distribution shift is the primary cause of the observed alteration in the loss function, resulting in the emergence of new minima. Gradient descent could potentially converge to a bad minimum, in which case the performance is significantly affected. To illustrate this intuition, we report in Fig. 10 the observed scaling laws for three different synthetic datasets. Let $N_{train} = 10^6$, $N_{test} = 3 \cdot 10^4$, $d = 1000$, and $m = 100$. The datasets are generated as follows:

1. *Linear* dataset: we first generate a random vector $W \sim \mathcal{N}(0, d^{-1} I_d)$. Then, we generate $N_{train}$ training samples and $N_{test}$ test samples with the rule $y = \mathbb{1}_{\{W^\top x > 0\}}$, where $x \in \mathbb{R}^d$ is simulated from $\mathcal{N}(0, I_d)$.

2. *NonLinear* dataset (Non-linearity): we first generate a random matrix $W_{in} \sim \mathcal{N}(0, d^{-1} I_{d \times m}) \in \mathbb{R}^{d \times m}$ and a random vector $W_{out} \sim \mathcal{N}(0, m^{-1} I_m)$. The samples are then generated with the rule $y = \mathbb{1}_{\{W_{out}^\top \phi(W_{in}^\top x)\}}$, where $x \in \mathbb{R}^d$ is simulated from $\mathcal{N}(0, I_d)$, and $\phi$ is the ReLU activation function. [9]

3. *NonLinear+Noisy* dataset: we first generate a random vector $W \sim \mathcal{N}(0, d^{-1} I_d)$. Then, we generate $N_{train}$ training samples and $N_{test}$ test samples with the rule $y = \mathbb{1}_{\{\sin(W^\top x + 0.3\epsilon) > 0\}}$, where

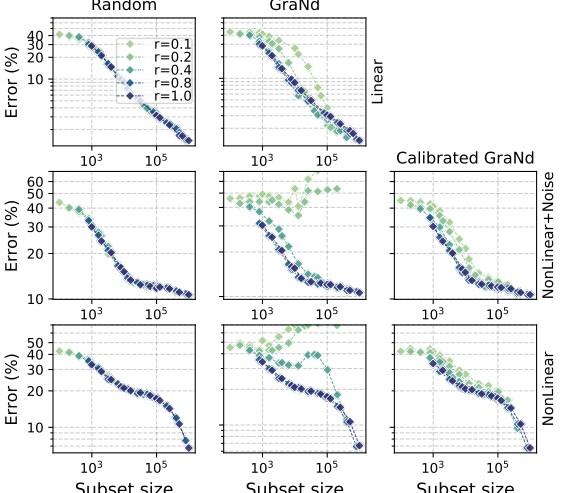

Figure 10: Test error on a 3-layers MLP (details are provided in Appendix D) on different pruned datasets for compression levels $r \in \{0.1, 0.2, 0.4, 0.8, 1\}$ where the pruning procedure is performed with `Random` pruning or `GraNd`. The case $r = 1$ corresponds to no pruning. In all the experiments, the network is trained until convergence.

$x \in \mathbb{R}^d$ is simulated from $\mathcal{N}(0, I_d)$ and $\epsilon$ is simulated from $\mathcal{N}(0, 1)$ and 'sin' refers to the sine function.

In Fig. 10, we compare the test error of an 3-layers MLP trained on different subsets generated with either `Random` pruning, or `GraNd`. As expected, with random pruning, the results are consistent regardless of the compression level $r$ as long as the subset size is the same. With `GraNd` however, the results depend on the difficulty of the dataset. For the linear dataset, it appears that we can indeed beat the power law scaling, provided that we have access to enough data. In contrast, `GraNd` seems to perform poorly on the nonlinear and noisy datasets in the high compression regime. This is due to the emergence of new local (bad) minima as $r$ decreases as evidenced in Fig. 1. Calibrated with the exact protocol, `GraNd` becomes valid: we can see that at any compression rate, the error converges to its minimum, which was not the case for $r \leq 20\%$. Whether calibration protocols can allow data pruning algorithms to beat the power law scaling remains however an open question: further research is needed in this direction. It is also worth noting that for the Nonlinear datasets, the scaling law pattern exhibits multi-phase behavior. For instance, for the *Nonlinear+Noisy* dataset, we can (visually) identify two phases, each one of which follows a different power law scaling pattern.

## 5.3 Image Tasks

Through our theoretical analysis, we have concluded that `SBPA` algorithms are generally non-consistent. This effect is most pronounced when the compression level $r$ is small. In this case, the loss landscape can be significantly altered due to the change in the data distribution caused by the pruning procedure. Given a `SBPA` algorithm, we argue that this alteration in distribution will inevitably affect the performance of the model trained on the pruned subset, and for small $r$, `Random` pruning becomes more effective than the `SBPA` algorithm.

---

[9]The ReLU activation function is given by $\phi(z) = \max(z, 0)$ for $z \in \mathbb{R}$. Here, we abuse the notation a bit and write $\phi(z) = (\phi(z_1), \ldots, \phi(z_m))$ for $z = (z_1, \ldots, z_m) \in \mathbb{R}^m$.

In the following, we empirically investigate this behaviour. We evaluate the performance of different `SBPA` algorithms from the literature and confirm our theoretical predictions with empirical evidence. We consider the following `SBPA` algorithms:

- `GraNd` (Paul, Ganguli, and Dziugaite, 2021): with this method, given a datapoint $z = (x, y)$, the score function $g$ is given by $g(z) = -\mathbb{E}_{w_t} \|\nabla_w \ell(y_{out}(x, w_t), y)\|^2$, where $y_{out}$ is the model output and $w_t$ are the model parameters (e.g. the weights in a neural network) at training step $t$, and where the expectation is taken with respect to random initialization. `GraNd` selects datapoints with the highest average gradient norm (w.r.t to initialization).

- `Uncertainty` (**Coleman2020Uncertainty**): in this method, the score function is designed to capture the uncertainty of the model in assigning a classification label to a given datapoint[10]. Different metrics can be used to measure this assignment uncertainty. We focus here on the entropy approach in which case the score function $g$ is given by $g(z) = \sum_{i=1}^{C} p_i(x) \log(p_i(x))$ where $p_i(x)$ is the model output probability that $x$ belongs to class $i$. For instance, in the context of neural networks, we have $(p_i(x))_{1 \leq i \leq C} = \texttt{Softmax}(y_{out}(x, w_t))$, where $t$ is the training step where data pruning is performed.

- `DeepFool` (Ducoffe and Precioso, 2018): this method is rooted in the idea that in a classification problem, data points that are nearest to the decision boundary are, in principle, the most valuable for the training process. While a closed-form expression of the margin is typically not available, the authors use a heuristic from the literature on adversarial attacks to estimate the distance to the boundary. Specifically, given a datapoint $z = (x, y)$, perturbations are added to the input $x$ until the model assigns the perturbed input to a different class. The amount of perturbation required to change the label for each datapoint defines the score function in this case (see (Ducoffe and Precioso, 2018) for more details).

We illustrate the limitations of the `SBPA` algorithms above for small $r$, and show that random pruning remains a strong baseline in this case. We further evaluate the performance of our calibration protocols and show that the signal parameter $\alpha$ can be tuned so that the calibrated `SBPA` algorithms outperform random pruning for small $r$. We conduct our experiments using the following setup:

- **Datasets and architectures.** Our framework is not constrained by the type of the learning task or the model. However, for our empirical evaluations, we focus on classification tasks with neural network models. We consider two image datasets: `CIFAR10` with `ResNet18` and `CIFAR100` with `ResNet34`. More datasets and neural architectures are available in our code, which is based on that of Guo, B. Zhao, and Bai, 2022. The code to reproduce all our experiments will be soon open-sourced.

- **Training.** We train all models using SGD with a decaying learning rate schedule that was empirically selected following a grid search. This learning rate schedule was also used in Guo, B. Zhao, and Bai, 2022. More details are provided in Appendix D.

- **Selection epoch.** The selection of the coreset can be performed at differnt training stages. We consider data pruning at two different training epochs: 1, and 5. We found that going beyond epoch 5 (e.g., using a selection epoch of 10) has minimal impact on the performance as compared to using a selection epoch of 5.

- **Pruning methods.** We consider the following data pruning methods: `Random, GraNd, DeepFool, Uncertainty`. In addition, we consider the pruning methods resulting from applying the proposed exact calibration protocol to a given `SBPA` algorithm. We use the notation `SBPA-CP1` to refer to the resulting method. For instance, `DeepFool-CP1` refers to the method resulting from applying (EC) to `DeepFool`.

---

[10]`Uncertainty` is specifically designed to be used for classification tasks. This means that it is not well-suited for other types of tasks, such as regression.

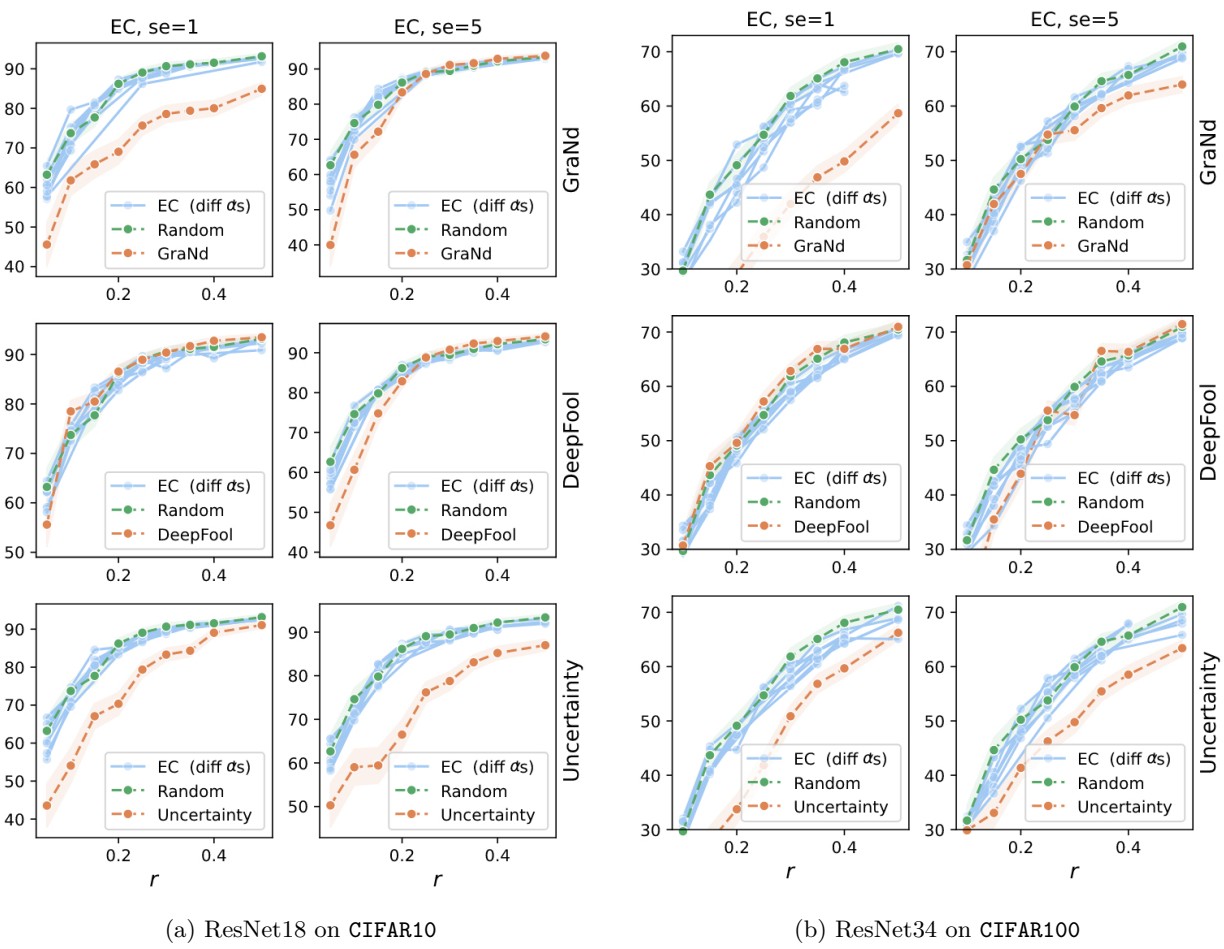

(a) ResNet18 on `CIFAR10`  (b) ResNet34 on `CIFAR100`

Figure 11: Test accuracy for different pruning methods, fractions $r$, signal parameters $\alpha$, and selection epochs ($se = 1$ or 5). Confidence intervals based on 3 runs are shown.

**Poor performance of `SBPA` in the high compression regime:** Fig. 11 shows the results of the data pruning methods described above with `ResNet18` on `CIFAR10` and `ResNet34` on `CIFAR100`. As expected, we observe a consistent decline in the performance of the trained model when the compression ratio $r$ is small, typically in the region $r < 0.3$. More importantly, we observe that `SBPA` methods (`GraNd`, `DeepFool`, `Uncertainty` in orange) perform consistently worse than `Random` pruning (in green), confirming our hypothesis. We also observe that amongst the three `SBPA` methods, `DeepFool` is generally the best in the region of interest of $r$ and competes with random pruning when the subset selection is performed at training epoch 1. We noticed that in that setting `DeepFool` is close to random pruning.

**Effect of the calibration protocol** Our proposed calibration protocol aim to correct the bias by injecting some randomness in the selection process and keeping (on average) only a fraction $\alpha$ of the `SBPA` method. We notice that the calibration protocol applied to different `SBPA` consistently boosts the performance in the high compression regime, as can be observed in Fig. 11. Fig. 12 shows that the calibrated `SBPA` perform better than `Random` pruning for specific choices of $\alpha$. However, the difference is not always significant. Besides, finding the optimal proportion of signal $\alpha$ can be difficult in practice.

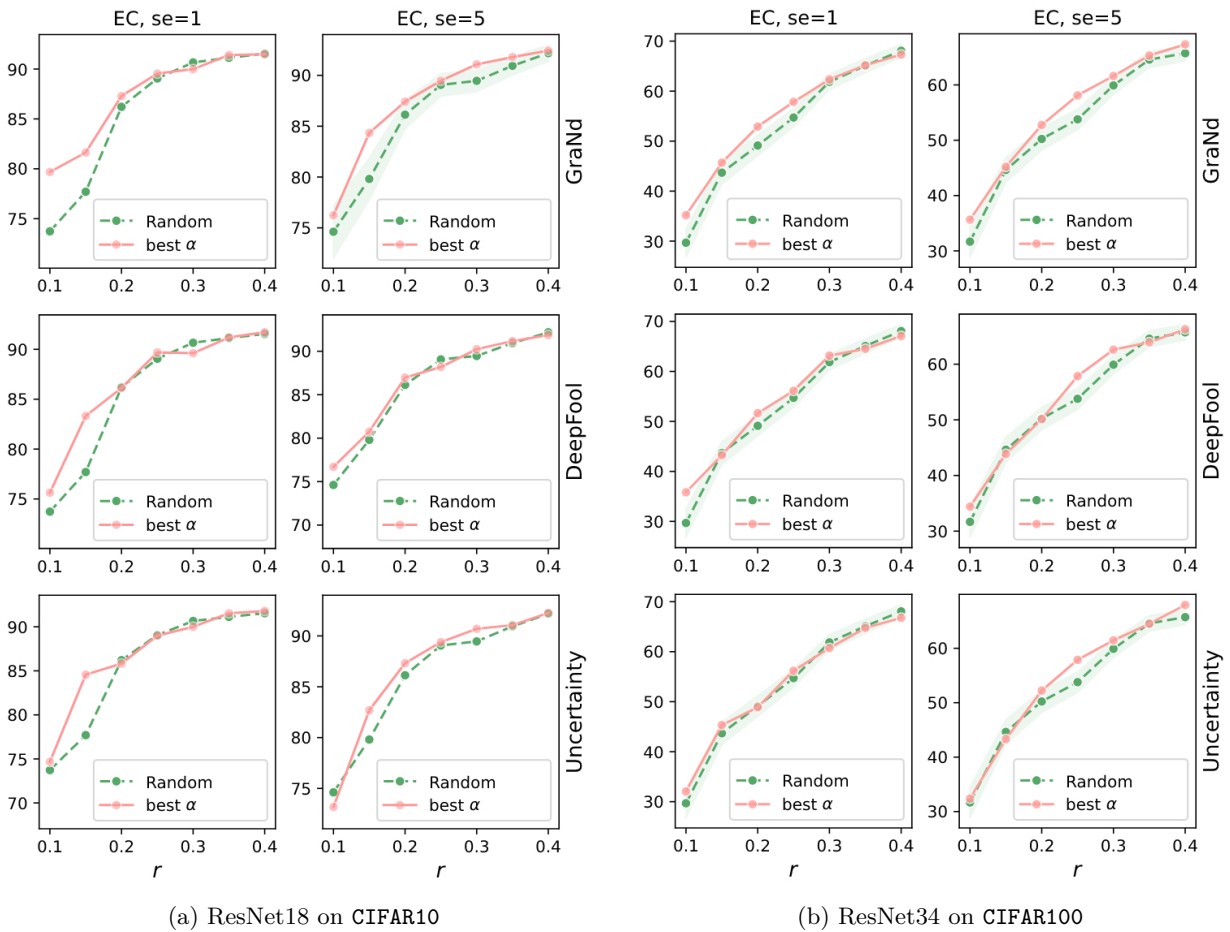

Figure 12: Test accuracy for different pruning methods, fractions $r$, and selection epochs ($se = 1$ or $5$). Best $\alpha$ used for calibration. Different values of $r$ may have different $\alpha$ values.

# 6 Related Work

As we mentioned in the introduction. The topic of coreset selection has been extensively studied in classical machine learning and statistics (Welling, 2009; Chen, Welling, and Smola, 2012; Feldman, Faulkner, and Krause, 2011; Huggins, Campbell, and Broderick, 2016; Campbell and Broderick, 2019). These classical approaches were either model-independent or designed for simple models (e.g. linear models). The recent advances in deep learning has motivated the need for new adapted methods for these deep models. Many approaches have been proposed to adapt to the challenges of the deep learning context. We will cover existing methods that are part of our framework (`SBPA` algorithms) and others that fall under different frameworks (non-`SBPA` algorithms).

## 6.1 Score-based Methods

These can generally be categorized into four groups:

1. Geometry based methods: these methods are based on some geometric measure in the feature space. The idea is to remove redundant examples in this feature space (examples that similar representations). Examples include Herding ((Chen, Welling, and Smola, 2012)) which aims to greedily select examples by ensuring that the centers of the coreset and that of the full dataset are

close. A similar idea based on the K-centroids of the input data was used in (Sener and Savarese, 2018; Agarwal et al., 2020; Sinha et al., 2020).

2. Uncertainty based methods: the aim of such methods is to find the most "difficult" examples, defined as the ones for which the model is the least confident. Different uncertainty measures can be used for this purpose, see (Coleman et al., 2020) for more details.

3. Error based methods: the goal is to find the most significant examples defined as the ones that contribute the most to the loss. In Paul, Ganguli, and Dziugaite, 2021, the authors consider the second norm of the gradient as a proxy to find such examples. Indeed, examples with the highest gradient norm tends to affect the loss more significantly (a first order Taylor expansion of the loss function can explain the intuition behind this proxy). This can be thought of as a relaxation of a Lipschitz-constant based pruning algorithm that was recently introduced in Ayed and Hayou, 2022. Another method consider keeping the most forgettable examples defined as those that change the most often from being well classified to being mis-classified during the course of the training (Toneva et al., 2019). Other methods in this direction consider a score function based on the relative contribution of each example to the total loss over all training examples (see Bachem, Lucic, and Krause, 2015; Munteanu et al., 2021).

4. Decision boundary based: although this can be encapsulated in uncertainty-based methods, the idea behind these methods is more specific. The aim is to find the examples near the decision boundary, the points for which the prediction has the highest variation (e.g. with respect to the input space, Ducoffe and Precioso, 2018; Margatina et al., 2021).

## 6.2 Non-SBPA Methods

Other methods in the literature select the coreset based on other desirable properties. For instance, one could argue that preserving the gradient is an important feature to have in the coreset as it would lead to similar minima (Killamsetty, Sivasubramanian, Ramakrishnan, De, et al., 2021; Mirzasoleiman, Bilmes, and Leskovec, 2020). Other work considered the problem of corset selection as a two-stage optimization problem where the subset selection can be seen also as an optimization problem (Killamsetty, Sivasubramanian, Ramakrishnan, and Iyer, 2021; Killamsetty, X. Zhao, et al., 2021). Other methods consider conisder the likelihood and its connection with submodular functions in order to select the subset (**kaushal2021prism**; Kothawade et al., 2021).

It is worth noting that there exist other approaches to data pruning that involve synthesizing a new dataset with smaller size that preserves certain desired properties, often through the brute-force construction of samples that may not necessarily represent the original data. These methods are known as data distillation methods (see e.g. Wang et al., 2020; B. Zhao, Mopuri, and Bilen, 2021; B. Zhao and Bilen, 2021) However, these methods have significant limitations, including the difficulty of interpreting the synthesized samples and the significant computational cost. The interpretability issue is particularly a these approaches to use in real-world applications, particularly in high-stakes fields such as medicine and financial engineering.

# 7 Discussion and Limitations

## 7.1 Extreme Scenarios

Our framework provides insights in the case where both $n$ and $rn$ are large. As a result, there are cases where this framework is not applicable. We call these cases extreme scenarios.

**Extreme scenario 1: small $n$.** Our asymptotic analysis can provide insights when a sufficient number of samples are available. In the scarce data regime (small $n$), our theoretical results may not accurately reflect the impact of pruning on the loss function. It is worth noting, however, that this case is generally not of practical interest as there is no benefit to data pruning when the sample size is small.

**Extreme scenario 2: large $n$ with $r = \Theta(n^{-1})$).** In this case, the "effective" sample size after pruning is $r\,n = \Theta(1)$. Therefore, we cannot glean useful information from the asymptotic behavior of $\mathcal{L}_n^{\mathcal{A},r}$ in this case. It is also worth noting that the variance of $\mathcal{L}_n^{\mathcal{A},r}$ does not vanish in the limit $n \to \infty, r \to 0$ with $rn = \gamma$ fixed, and therefore the empirical mean does not converge to the asymptotic mean.

## 7.2 Asymptotic Results

Theorem 1 and 2, and the subsequent corollaries are asymptotic results. They essentially reveal the limitations of `SBPA` for "large enough models". We decided to take this direction to get results that are as general as possible, showing that the discussed limitations will appear in most situations. Theorem 1 and 2 apply to any configuration from a large variety of classes of models, SBPAs, generating processes, and loss functions. The theory readily covers realistic architectures illustrated by Corollary 3, 4 and 5 (in the appendix) that cover wide NN, deep neural NN, and convolutional NN with enough filters. However, these limitations could appear for unrealistically large models. We acknowledge this drawback of the proposed theory and address it in two ways. First, we experimentally show that for the usual settings (ResNet on Cifar), we already observe this significant drop in the performance of SBPAs compared to random pruning. This also aligns with other empirical observations (Guo, B. Zhao, and Bai, 2022). In addition, we provide Theorem 3 to cover the cases where the class of functions is potentially not rich enough; even in that case, for any SBPA, one can find datasets for which the pruning algorithm will fail for small compression ratios. Besides, to our knowledge, the lines of work that allow one to derive explicit bounds for Neural Networks usually require specific and often overly simplistic architectures (typically one hidden layer feed-forward). In our context, we additionally expect similar strong restrictions to be required for the SBPAs and generating processes. This could wrongfully lead practitioners to consider that using a different `SBPA` or class of models than the one for which one could derive quantitative results would circumvent the limitations.

## 7.3 Overparameterized Regime

The scenario in which both the number of parameters $p$ and the sample size $n$ tend to infinity (e.g. with a constant ratio $\gamma = p/n$) holds practical significance. Our framework does not cover this case and we would like to elucidate the main point at which our proof machinery encounters challenges under this scenario. The main issue resides in understanding the asymptotic behaviour of SBPA algorithms in this context, particularly the extension of Proposition 3. While we can establish concentration with $p$ constant and $n$ growing large, achieving this with both $n$ and $p$ going to infinity, especially when the underlying model is a neural network, becomes generally intractable. Nonetheless, under certain supplementary assumptions, it remains feasible to demonstrate concentration.

Recall that in Proposition 3, we essentially use a variation of the law of large numbers to show the convergence in the infinite sample size limit $(n \to \infty)$, while $p$ (and consequently, $w$) is fixed. However, in the scenario where both $p$ and $n$ tend to infinity, the dependency of $w$ on $p$ complicates matters, rendering the used variation of the LLN inapplicable. An essential condition under such circumstances becomes the convergence of $w$ in a certain sense as well, as $p \to \infty$. For this purpose, a pertinent tool is LLN for Triangular Arrays, which takes the shape of:

*Consider a Triangular Array $(X_{n,i})_{i \in [n], n \geq 1}$ of random variables such that for each $n$, the variables $(X_{n,i})_{i \in [n]}$ are iid with mean $\mu_n$. Then, under some assumptions (bounded second moments) we have*

$$n^{-1}\left(\sum_{i=1}^{n} X_{n,i} - \mu_n\right) \to \infty.$$

However, note that in our case, the terms $X_{n,i} = \ell(y_{out}(X_i; w_p), Y_i)\mathbf{1}_{\{i \text{ is accepted}\}}$ are not necessarily independent, and therefore more advanced LLN for trinagular arrays should be proven and used. We leave this for future work.

### 7.4 Pruning Time Vs Training Time

In some cases, the pruning procedure might be compute-intensive, and requires more resources than the actual training with the full dataset. This is the case when, for instance, the pruning criterion depends on second order geometry (Hessian etc) and/or multiple perturbations of some quantity (DeepFool), or pruning is performed in multi-shot settings (dynamical pruning). In this paper, we considered pruning criteria that can be performed "one shot" and rely on criteria that involve either gradients (GraNd) or network outputs (Uncertainty) or perturbations of the network output (DeepFool). For GraNd and Uncertainty pruning, the pruning time is typically less than the time required for 1 epoch of training. For DeepFool however, the pruning time might be comparable to 10 epochs of training.

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

# 8 Acknowledgement

We would like to thank the authors of DeepCore project (Guo, B. Zhao, and Bai, 2022) for open-sourcing their excellent code[11]. The high flexibility and modularity of their code allowed us to quickly implement our calibration protocols on top of existing SBPA algorithms.

---

[11]The code by Guo, B. Zhao, and Bai, 2022 is available at `https://github.com/PatrickZH/DeepCore`.

# A  Proofs

## A.1  Proofs of Section 2

Propositions 1 and 2 are built on the following lemma.

**Lemma 1** *Let $\pi$ be a distribution on $\mathcal{D}$ and $(w_n)_n$ a sequence of parameters in $\mathcal{W}_\theta$ satisfying*

$$\mathbb{E}_\pi \ell(y_{out}(X; w_n), Y) \to \min_{w \in \mathcal{W}_\theta} \mathbb{E}_\pi \ell(y_{out}(X; w), Y).$$

*Then, it comes that*

$$d(w_n, \mathcal{W}_\theta^*(\pi)) \to 0.$$

**Proof:**  Denote $\mathcal{L}_\pi$ the function from $\mathcal{W}_\theta$ to $\mathbb{R}$ defined by

$$\mathcal{L}_\pi(w) = \mathbb{E}_\pi \ell(y_{out}(X; w), Y).$$

Notice that under our assumptions, the dominated convergence theorem gives that $\mathcal{L}_\pi$ is continuous. This lemma is a simple consequence of the continuity of $\mathcal{L}_\pi$ and the compacity of $\mathcal{W}_\theta$. Consider a sequence $(w_n)$ such that

$$\mathcal{L}_\pi(w_n) \to \min_{w \in \mathcal{W}_\theta} \mathcal{L}_\pi(w).$$

We can prove the lemma by contradiction. Consider $\epsilon > 0$ and assume that there exists infinitely many indices $n_k$ for which $d\left(w_{n_k}, \mathcal{W}_\theta^*(\pi)\right) > \epsilon$. Since $\mathcal{W}_\theta$ is compact, we can assume that $w_{n_k}$ is convergent (by considering a subsequence of which if needed), denote $w_\infty \in \mathcal{W}_\theta$ its limit. The continuity of $d$ then gives that $d\left(w_\infty, \mathcal{W}_\theta^*(\pi)\right) \geq \epsilon$, and in particular

$$w_\infty \notin \mathcal{W}_\theta^*(\pi) = \operatorname{argmin}_{w \in \mathcal{W}_\theta} \mathcal{L}_\pi(w).$$

But since $\mathcal{L}_\pi$ is continuous, the initial assumption on $(w_n)$ translates to

$$\min_{w \in \mathcal{W}_\theta} \mathcal{L}_\pi(w) = \lim_k \mathcal{L}_\pi(w_{n_k}) = \mathcal{L}_\pi(w_\infty),$$

concluding the proof. $\qquad\square$

**Proposition 1.** *A pruning algorithm $\mathcal{A}$ is valid at a compression ratio $r \in (0, 1]$ if and only if*

$$d\left(w_n^{\mathcal{A}, r}, \mathcal{W}_\theta^*(\mu)\right) \to 0 \ a.s.$$

*where $\mathcal{W}_\theta^*(\mu) = \operatorname{argmin}_{w \in \mathcal{W}_\theta} \mathcal{L}(w) \subset \mathcal{W}_\theta$ and $d\left(w_n^{\mathcal{A}, r}, \mathcal{W}_\theta^*(\mu)\right)$ denotes the euclidean distance from the point $w_n^{\mathcal{A}, r}$ to the set $\mathcal{W}_\theta^*(\mu)$.*

**Proof:**  This proposition is a direct consequence of Lemma 1. Consider a valid pruning algorithm $\mathcal{A}$, a compression ratio $r$ and a sequence of observations $(X_k, Y_k)$ such that

$$\mathcal{L}(w_n^{\mathcal{A}, r}) \to \min_{w \in \mathcal{W}_\theta} \mathcal{L}(w).$$

We can apply Lemma 1 on the sequence $(w_n^{\mathcal{A}, r})$ with the distribution $\pi = \mu$ to get the result. $\qquad\square$

**Proposition 2.** *Let $\mathcal{A}$ be a pruning algorithm and $r \in (0, 1]$ a compression ratio. Assume that there exists a probability measure $\nu_r$ on $\mathcal{D}$ such that*

$$\forall w \in \mathcal{W}_\theta, \ \mathcal{L}_n^{\mathcal{A}, r}(w) \to \mathbb{E}_{\nu_r} \ell(y_{out}(X; w), Y) \ a.s. \tag{5}$$

*Then, denoting $\mathcal{W}_\theta^*(\nu_r) = \operatorname{argmin}_{w \in \mathcal{W}_\theta} \mathbb{E}_{\nu_r} \ell(y_{out}(X; w), Y) \subset \mathcal{W}_\theta$, we have that*

$$d\left(w_n^{\mathcal{A}, r}, \mathcal{W}_\theta^*(\nu_r)\right) \to 0 \ a.s.$$

**Proof:** Leveraging Lemma 1, it is enough to prove that

$$\mathbb{E}_{\nu_r}\ell(y_{out}(X; w_n^{\mathcal{A},r}), Y) - \min_{w \in \mathcal{W}_\theta} \mathbb{E}_{\nu_r}\ell(y_{out}(X; w), Y) \to 0 \ a.s.$$

To simplify the notations, we introduce the function $f$ from $\mathcal{D} \times \mathcal{W}_\theta$ to $\mathbb{R}$ defined by

$$f(z, w) = \ell(y_{out}(x; w), y),$$

where $z = (x, y)$. Since $\mathcal{W}_\theta$ is compact, we can find $w^* \in \mathcal{W}_\theta$ such that $\mathbb{E}_{\nu_r}[f(z, w^*)] = \min_w \mathbb{E}_{\nu_r}[f(z, w)]$. It comes that

$$
\begin{aligned}
0 \ &\leq \ \mathbb{E}_{\nu_r}[f(z, w_n^{\mathcal{A},r})] - \mathbb{E}_{\nu_r}[f(z, w^*)] \\
&\leq \ \mathbb{E}_{\nu_r}[f(z, w_n^{\mathcal{A},r})] - \frac{1}{rn} \sum_{z \in \mathcal{A}(\mathcal{D}_n, r)} f(z, w_n^{\mathcal{A},r}) \\
&+ \ \frac{1}{rn} \sum_{z \in \mathcal{A}(\mathcal{D}_n, r)} f(z, w_n^{\mathcal{A},r}) - \frac{1}{rn} \sum_{z \in \mathcal{A}(\mathcal{D}_n, r)} f(z, w^*) \\
&+ \ \frac{1}{rn} \sum_{z \in \mathcal{A}(\mathcal{D}_n, r)} f(z, w^*) - \mathbb{E}_{\nu_r}[f(z, w^*)]
\end{aligned}
$$

The last term converges to zero almost surely by assumption. By definition of $w_n^{\mathcal{A},r}$, the middle term is non-positive. It remains to show that the first term also converges to zero. With this, we can conclude that $\lim_n \mathbb{E}_{\nu_r}[f(z, w_n^{\mathcal{A},r})] - \mathbb{E}_{\nu_r}[f(z, w^*)] = 0$

To prove that the first term converges to zero, we use the classical result that if every subsequence of a sequence $(u_n)$ has a further subsequence that converges to $u$, then the sequence $(u_n)$ converges to $u$. Denote

$$u_n = \mathbb{E}_{\nu_r}[f(z, w_n^{\mathcal{A},r})] - \frac{1}{rn} \sum_{z \in \mathcal{A}(\mathcal{D}_n, r)} f(z, w_n^{\mathcal{A},r}).$$

By compacity of $\mathcal{W}_\theta$, from any subsequence of $(u_n)$ we can extract a further subsequence with indices denoted $(n_k)$ such that $w_{n_k}^*$ converges to some $w_\infty \in \mathcal{W}_\theta$. We will show that $(u_{n_k})$ converges to 0. Let $\epsilon > 0$, since $f$ is continuous on the compact set $\mathcal{D} \times \mathcal{W}_\theta$, it is uniformly continuous. Therefore, almost surely, for $k$ large enough,

$$\sup_z |f(z, w_{n_k}^*) - f(z, w_\infty)| \leq \epsilon.$$

Denoting

$$v_n = \mathbb{E}_{\nu_r}[f(z, w_\infty)] - \frac{1}{rn} \sum_{z \in \mathcal{A}(\mathcal{D}_n, r)} f(z, w_\infty),$$

the triangular inequality then gives that, almost surely, for $k$ large enough

$$|u_{n_k} - v_{n_k}| \leq 2\epsilon.$$

By assumption, the sequence $v_{n_k}$ converges to zero almost surely, which concludes the proof. $\qquad \square$

We now prove Corollary 2, since Corollary 1 is a straightforward application of Proposition 2.

**Corollary 2.** *Let $\mathcal{A}$ be any pruning algorithm and $r \in (0, 1]$, and assume that (5) holds for a given probability measure $\nu_r$ on $\mathcal{D}$. If $\mathcal{A}$ is valid, then $\mathcal{W}_\theta^*(\nu_r) \cap \mathcal{W}_\theta^*(\mu) \neq \emptyset$; or, equivalently,*

$$\min_{w \in \mathcal{W}_\theta^*(\nu_r)} \mathcal{L}(w) = \min_{w \in \mathcal{W}} \mathcal{L}(w).$$

**Proof:** This proposition is a direct consequence of Proposition 2 that states that

$$d(w_n^{\mathcal{A},r}, \mathcal{W}_\theta^*(\nu_r)) \to 0 \ a.s.$$

Since the $\mathcal{L}$ is continuous on the compact $\mathcal{W}_\theta$, it is uniformly continuous. Hence, for any $\epsilon > 0$, we can find $\eta > 0$ such that if $d(w, w') \leq \eta$, then $|\mathcal{L}(w) - \mathcal{L}(w')| \leq \epsilon$ for any parameters $w, w' \in \mathcal{W}_\theta$. Hence, for $n$ large enough, $d(w_n^{\mathcal{A},r}, \mathcal{W}_\theta^*(\nu_r)) \leq \eta$, leading to

$$\mathcal{L}(w_n^{\mathcal{A},r}) \geq \min_{w \in \mathcal{W}_\theta^*(r)} \mathcal{L}(w) - \epsilon.$$

Since the algorithm is valid, we know that $\mathcal{L}(w_n^{\mathcal{A},r})$ converges to $\min_{w \in \mathcal{W}_\theta} \mathcal{L}(w)$ almost surely. Therefore, for any $\epsilon > 0$,

$$\min_{w \in \mathcal{W}_\theta} \mathcal{L}(w) \geq \min_{w \in \mathcal{W}_\theta^*(r)} \mathcal{L}(w) - \epsilon.$$

which concludes the proof. $\qquad\square$

### A.2 Proof of Proposition 3

**Proposition 3.** [Asymptotic behavior of `SBPA`]
*Let $\mathcal{A}$ be a `SBPA` algorithm and let $g$ be its corresponding score function. Assume that $g$ is adapted, and consider a compression ratio $r \in (0,1)$. Denote by $q^r$ the $r^{th}$ quantile of the random variable $g(Z)$ where $Z \sim \mu$. Denote $A_r = \{z \in \mathcal{D} \mid g(z) \leq q^r\}$. Almost surely, the empirical measure of the retained data samples converges weakly to $\nu_r = \frac{1}{r}\mu_{|A_r}$, where $\mu_{|A_r}$ is the restriction of $\mu$ to the set $A_r$. In particular, we have that*

$$\forall w \in \mathcal{W}_\theta, \ \mathcal{L}_n^{\mathcal{A},r}(w) \to \mathbb{E}_{\nu_r}\ell(y_{out}(X; w), Y) \ a.s.$$

**Proof:** Consider $\mathcal{F}$ the set of functions $f : \mathcal{D} \to [-1, 1]$ that are continuous. We will show that

$$\sup_{f \in \mathcal{F}} \left| \frac{1}{|\mathcal{A}(\mathcal{D}_n, r)|} \sum_{z \in \mathcal{A}(\mathcal{D}_n, r)} f(z) - \frac{1}{r} \int_{A_r} f(z)\mu(z)dz \right| \to 0 \ a.s. \qquad (9)$$

To simplify the notations, and since $\frac{|\mathcal{A}(\mathcal{D}_n, r)|}{rn}$ converges to 1, we will assume that $rn$ is an integer. Denote $q_n^r$ the $(rn)^{th}$ ordered statistic of $\left(g(z_i)\right)_{i=1,\ldots,n}$, and $q^r$ the $r^{th}$ quantile of the random variable $g(Z)$ where $Z \sim \mu$.

We can upper bound the left hand side in equation (9) by the sum of two random terms $A_n$ and $B_n$ defined by

$$B_n = \frac{1}{r} \sup_{f \in \mathcal{F}} \left| \frac{1}{n} \sum_{z \in \mathcal{D}_n} f(z)\mathbb{I}_{g(z) \leq q_n^r} - \frac{1}{n} \sum_{z \in \mathcal{D}_n} f(z)\mathbb{I}_{g(z) \leq q^r} \right|$$

$$C_n = \frac{1}{r} \sup_{f \in \mathcal{F}} \left| \frac{1}{n} \sum_{z \in \mathcal{D}_n} f(z)\mathbb{I}_{g(z) \leq q^r} - \int f(z)\mathbb{I}_{g(z) \leq q^r}\mu(z)dz \right|$$

To conclude the proof, we will show that both terms converge to zero almost surely.

For any $f \in \mathcal{F}$, denoting $G_n$ the empirical cumulative density function (cdf) of $(g(z_i))$ and $G$ the cdf of $g(Z)$, we have that

$$
\begin{aligned}
\left| \frac{1}{n} \sum_{z \in \mathcal{D}_n} f(z) \mathbb{I}_{g(z) \leq q_n^r} - \frac{1}{n} \sum_{z \in \mathcal{D}_n} f(z) \mathbb{I}_{g(z) \leq q^r} \right| 
&\leq \frac{1}{n} \sum_{z \in \mathcal{D}_n} |f(z)| \times \left| \mathbb{I}_{g(z) \leq q_n^r} - \mathbb{I}_{g(z) \leq q^r} \right| \\
&\leq \frac{1}{n} \sum_{z \in \mathcal{D}_n} \left| \mathbb{I}_{g(z) \leq q_n^r} - \mathbb{I}_{g(z) \leq q^r} \right| \\
&\leq |G_n(q_n^r) - G_n(q^r)| \\
&= \left| \frac{1}{r} - G_n(q^r) \right| \\
&= |G(q^r) - G_n(q^r)|.
\end{aligned}
$$

Therefore, $B_n \leq \sup_{t \in \mathbf{R}} |G(t) - G_n(t)|$ which converges to zero almost surely by the Glivenko-Cantelli theorem.

Similarly, the general Glivenko-Cantelli theorem for metric spaces (Varadarajan, 1958) gives that almost surely,

$$
\sup_{f \in \mathcal{F}} \left| \frac{1}{n} \sum_{z \in \mathcal{D}_n} f(z) - \int f(z) \mu(z) dz \right| \to 0.
$$

Consider $k \geq 1$. Since $g$ is continuous and $\mathcal{D}$ is compact, the sets $A_{r(1-1/k)}$ and $\overline{A_r} = \mathcal{D} \setminus A_r$ are disjoint and closed subsets. Using Urysohn's lemma (Theorem 8 in the Appendix), we can find $f_k \in \mathcal{F}$ such that $f_k(z) = 1$ if $z \in A_{r(1-1/k)}$ and $f_k(z) = 0$ if $z \in \overline{A_r}$. Consider $f \in \mathcal{F}$, it comes that

$$
\begin{aligned}
\left| \frac{1}{n} \sum_{z \in \mathcal{D}_n} f(z) \mathbb{I}_{g(z) \leq q^r} - \int f(z) \mathbb{I}_{g(z) \leq q^r} \mu(z) dz \right| 
&\leq \left| \frac{1}{n} \sum_{z \in \mathcal{D}_n} f \times f_k(z) - \int f \times f_k(z) \mu(z) dz \right| \\
&+ \frac{1}{n} \sum_{z \in \mathcal{D}_n} \mathbb{I}_{q^{r(1-1/k)} \leq g(z) \leq q^r} \\
&+ \int \mathbb{I}_{q^{r(1-1/k)} \leq g(z) \leq q^r} \mu(z) dz
\end{aligned}
$$

Hence, noticing that $f \times f_k \in \mathcal{F}$, we find that

$$
C_n \leq \sup_{f \in \mathcal{F}} \left| \frac{1}{n} \sum_{z \in \mathcal{D}_n} f(z) - \int f(z) \mu(z) dz \right| + |G_n(q^r) - G_n(q^{r(1-1/k)})| + \frac{r}{k}.
$$

We can conclude the proof by noticing that $|G_n(q^r) - G_n(q^{r(1-1/k)})|$ converges to $\frac{r}{k}$ and taking $k \to \infty$. $\square$

### A.3   Proof of Theorem 1

In order to prove the theorem, we will need a few technical results that we state and prove first.

**Lemma 2** *Consider a set of continuous functions $\mathcal{M}$ from $\mathcal{X}$ to $\mathcal{Y}$. Consider $\psi_0$ a function in the closure of $\mathcal{M}$ for the $\ell_\infty$ norm. Then for any $\epsilon > 0$, there exists $\psi \in \mathcal{M}$ such that*

$$
\sup_{x,y \in \mathcal{D}} \| \ell(\psi(x), y) - \ell(\psi_0(x), y) \| \leq \epsilon
$$

**Proof:**   Since the loss $\ell$ is continuous on the compact $\mathcal{Y} \times \mathcal{Y}$, it is uniformly continuous. We can therefore find $\eta > 0$ such that for any $y_0, y, y' \in \mathcal{Y}$, if $\|y - y'\| \leq \eta$ then $\|\ell(y_0, y) - \ell(y_0, y')\| \leq \epsilon$. We conclude the proof using by selecting any $\psi \in \mathcal{M}$ that is at a distance not larger than $\eta$ from $\psi_0$ for the $\ell_\infty$ norm.   $\square$

**Lemma 3** *Consider a* `SBPA` $\mathcal{A}$. *Let* $\mathcal{M}$ *be a set of continuous functions from* $\mathcal{X}$ *to* $\mathcal{Y}$. *Consider* $r \in (0,1)$ *and assume that* $\mathcal{A}$ *is consistent on* $\mathcal{M}$ *at level* $r$, *i.e.*

$$\forall \psi \in \mathcal{M}, \quad \frac{1}{|\mathcal{A}(\mathcal{D},r)|} \sum_{(x,y) \in \mathcal{A}(\mathcal{D},r)} \ell(\psi(x),y) \to \mathbb{E}_\mu \ell(\psi(X),Y) \ a.s.$$

*Let* $\psi_\infty$ *be any measurable function from* $\mathcal{X}$ *to* $\mathcal{Y}$. *If there exists a sequence of elements of* $\mathcal{M}$ *that converges point-wise to* $\psi_\infty$, *then*

$$\mathbb{E}_{\frac{1}{r}\mu_{|A_r}} \ell(\psi_\infty(X),Y) = \mathbb{E}_\mu \ell(\psi_\infty(X),Y). \tag{10}$$

*In particular, if* $\mathcal{M}$ *has the universal approximation property, then* (10) *holds for any continuous function.*

**Proof:** Le $(\psi_k)_k$ be a sequence of functions in $\mathcal{M}$ that converges point-wise to $\psi_\infty$. Consider $k \geq 0$, since $\mathcal{A}$ is consistent and that $\psi_k$ is continuous and bounded, Proposition 3 gives that

$$\mathbb{E}_{\frac{1}{r}\mu_{|A_r}} \ell(\psi_k(X),Y) = \mathbb{E}_\mu \ell(\psi_k(X),Y).$$

Since $\ell$ is bounded, we can apply the dominated convergence theorem to both sides of the equation to get the final result. $\qquad\square$

Proposition 5 hereafter proves the final result for `SBPA` that do not depend on the labels. The proof of Theorem 1 that follows essentially deals with the remaining case of `SBPA` that depend on the labels.

**Proposition 5** *Let* $\mathcal{A}$ *be any* `SBPA` *with an adapted score function* $g$ *satisfying*

$$\exists \tilde{g} : \mathcal{X} \to \mathbb{R}_+, \ g(x,y) = \tilde{g}(x) \ \ a.s.$$

*Assume that there exists two continuous functions* $f_1$ *and* $f_2$ *such that*

$$\mathbb{E}_\mu \ell(f_1(X),Y) \neq \mathbb{E}_\mu \ell(f_2(X),Y).$$

*If* $\cup_\theta \mathcal{M}_\theta$ *has the universal approximation property, then there exist hyper-parameters* $\theta \in \Theta$ *for which the algorithm is not consistent.*

**Proof:** Consider a compression ratio $r \in (0,1)$. We will prove the result by means of contradiction. Assume that the `SBPA` is consistent on $\cup_\theta \mathcal{M}_\theta$. From the universal approximation property and Lemma 3, we get that

$$\frac{1}{r}\mathbb{E}_{\mu_{|A_r}} \ell(f_1(X),Y) = \mathbb{E}_\mu \ell(f_1(X),Y),$$

from which we deduce that

$$\mathbb{E}_\mu \left[ \ell(f_1(X),Y)\,\mathbb{I}(Z \in A_r) \right] = r\, \mathbb{E}_\mu \ell(f_1(X),Y) \tag{11}$$

$$\mathbb{E}_\mu \left[ \ell(f_1(X),Y)\,\mathbb{I}(Z \in \mathcal{D} \setminus A_r) \right] = (1-r)\, \mathbb{E}_\mu \ell(f_1(X),Y) \tag{12}$$

and similarly for $f_2$.

Notice that since the score function $g$ does not depend on $Y$, there exists $\mathcal{X}_r \subset \mathcal{X}$ such that $A_r = \mathcal{X}_r \times \mathcal{Y}$. Consider the function defined by

$$f : x \mapsto f_1(x)\mathbb{I}(x \in \mathcal{X}_r) + f_2(x)\left(1 - \mathbb{I}(x \in \mathcal{X}_r)\right),$$

we will show that

i) $\frac{1}{r}\mathbb{E}_{\mu_{|A_r}} \ell(f(X),Y) \neq \mathbb{E}_\mu \ell(f(X),Y)$

ii) There exists a sequence of elements in $\cup_\theta \mathcal{M}_\theta$ that converges point-wise almost everywhere to $f$

The conjunction of these two points contradicts Lemma 3, which would conclude the proof.

The first point is obtained through simple derivations, evaluating both sides of the equation i).

$$
\begin{aligned}
\frac{1}{r}\mathbb{E}_{\mu_{|A_r}}\ell\left(f(X),Y\right) &= \frac{1}{r}\mathbb{E}_\mu\ell\left(f(X),Y\right)\mathbb{I}(Z \in \mathcal{X}_r \times \mathcal{Y}) \\
&= \frac{1}{r}\mathbb{E}_\mu\ell\left(f(X),Y\right)\mathbb{I}(X \in \mathcal{X}_r) \\
&= \frac{1}{r}\mathbb{E}_\mu\ell\left(f_1(X),Y\right)\mathbb{I}(X \in \mathcal{X}_r) \\
&= \frac{1}{r}\mathbb{E}_\mu\ell\left(f_1(X),Y\right)\mathbb{I}(Z \in A_r) \\
&= \mathbb{E}_\mu\ell\left(f_1(X),Y\right),
\end{aligned}
$$

where we successively used the definition of $f$ and equation (11). Now, using the definition of $f$, we get that

$$
\begin{aligned}
\mathbb{E}_\mu\ell\left(f(X),Y\right) &= \mathbb{E}_\mu\ell\left(f_1(X),Y\right)\mathbb{I}(X \in \mathcal{X}_r) + \mathbb{E}_\mu\ell\left(f_2(X),Y\right)(1 - \mathbb{I}(X \in \mathcal{X}_r)) \\
&= \mathbb{E}_\mu\ell\left(f_1(X),Y\right)\mathbb{I}(Z \in A_r) + \mathbb{E}_\mu\ell\left(f_2(X),Y\right)\mathbb{I}(Z \in \mathcal{D} \setminus A_r) \\
&= r\mathbb{E}_\mu\ell\left(f_1(X),Y\right) + (1-r)\mathbb{E}_\mu\ell\left(f_2(X),Y\right).
\end{aligned}
$$

These derivations lead to

$$
\frac{1}{r}\mathbb{E}_{\mu_{|A_r}}\ell\left(f(X),Y\right) - \mathbb{E}_\mu\ell\left(f(X),Y\right) = (1-r)\left[\mathbb{E}_\mu\ell\left(f_1(X),Y\right) - \mathbb{E}_\mu\ell\left(f_2(X),Y\right)\right] \neq 0,
$$

by assumption on $f_1$ and $f_2$.

For point ii), we will construct a sequence $(\psi_k)_k$ of functions in $\cup_\theta \mathcal{M}_\theta$ that converges point-wise to $f$ almost everywhere, using the definition of the universal approximation property and Urysohn's lemma (Lemma 8 in the Appendix). Consider $k \geq 0$ and denote $\epsilon_k = \frac{1-r}{k+1}$. Denote $q^r$ and $q^{r+\epsilon_k}$ the $r^{th}$ and $(r+\epsilon_k)^{th}$ quantile of the random variable $\tilde{g}(X)$ where $(X,Y) \sim \mu$. Denote $\mathcal{X}_r = \{x \in \mathcal{X} \mid \tilde{g}(x) \leq q^r\}$ and $B_{r,k} = \{x \in \mathcal{X} \mid \tilde{g}_r(x) \geq q^{r+\epsilon_k}\}$. Since $\tilde{g}$ is continuous and $\mathcal{X}$ is compact, the two sets are closed. Besides, since the random variable $\tilde{g}(X)$ is continuous ($g$ is an adapted score function), both sets are disjoint. Therefore, using Urysohn's lemma (Lemma 8 in the Appendix), we can chose a continuous function $\phi_k : \mathcal{X} \to [0,1]$ such that $\phi_k(x) = 1$ for $x \in \mathcal{X}_r$ and $\phi_k(x) = 0$ for $x \in B_{r,k}$. Denote $f_k$ the function defined by

$$
\bar{f}_k(x) = f_1(x)\phi_k(x) + f_2(x)(1 - \phi_k(x)).
$$

Notice that $(\phi_k)_k$ converges point-wise to $\mathbb{I}(\cdot \in \mathcal{X}_r)$, and therefore $(\bar{f}_k)_k$ converges point-wise to $f$. Besides, since $\bar{f}_k$ is continuous, and $\cup_\theta \mathcal{M}_\theta$ has the universal approximation property, we can chose $\psi_k \in \cup_\theta \mathcal{M}_\theta$ such that

$$
\sup_{x \in \mathcal{X}}|\psi_k(x) - \bar{f}_k(x)| \leq \epsilon_k.
$$

Hence, for any input $x \in \mathcal{X}$, we can upper-bound $|\psi_k(x) - f(x)|$ by $\epsilon_k + |\bar{f}_k(x) - f(x)|$, giving that $\psi_k$ converges pointwise to $f$ and concluding the proof. □

We are now ready to prove the Theorem 1 that we state here for convenience.

**Theorem 1.** *Let $\mathcal{A}$ be any `SBPA` algorithm with an adapted score function. If $\cup_\theta \mathcal{M}_\theta$ has the universal approximation property, then there exist hyper-parameters $\theta \in \Theta$ for which the algorithm is not consistent.*

**Proof:** We will use the universal approximation theorem to construct a model for which the algorithm is biased. Denote $\mathrm{supp}(\mu)$ the support of the generating measure $\mu$. We can assume that there exists $x \in \mathcal{X}$ such that $(x_0, 0) \in \mathrm{supp}(\mu)$, $(x_0, 1) \in \mathrm{supp}(\mu)$, and $g(x_0, 1) \neq g(x_0, 0)$, otherwise one can apply Proposition 5 to get the result. Denote $y_0 \in \{0,1\}$ such that $g(x_0, y_0) > g(x_0, 1 - y_0)$. Since $g$ is continuous, we can find $\epsilon > 0, r_0 \in (0,1)$ such that

$$
\forall x \in \mathcal{B}(x_0, \epsilon),\ g(x, y_0) > q^{r_0} > g(x, 1 - y_0), \tag{13}
$$

where $q^{r_0}$ is the $r_0^{th}$ quantile of $g(Z)$ where $Z \sim \mu$.

Since $(x_0, 1 - y_0) \in \text{supp}(\mu)$, it comes that

$$\Delta = \frac{1 - r_0}{2(1 + r_0)} \mathbb{P}\Big(X \in \mathcal{B}(x_0, \epsilon), Y = 1 - y_0\Big) \ell(y_0, 1 - y_0) > 0.$$

By assumption, the distribution of $X$ is dominated by the Lebesgue measure, we can therefore find a positive $\epsilon' < \epsilon$ such that

$$\mathbb{P}\Big(X \in \mathcal{B}(x_0, \epsilon) \setminus \mathcal{B}(x_0, \epsilon')\Big) < \frac{\Delta}{2 \max \ell}.$$

The sets $K_1 = \mathcal{B}(x_0, \epsilon')$ and $K_2 = \mathcal{X} \setminus \mathcal{B}_o(x_0, \epsilon)$ are closed and disjoint sets, Lemma 8 in Appendix insures the existance of a continuous function $h$ such that $h(x) = y_0$ for $x \in K_1$, and $h(x) = 1 - y_0$ for $x \in K_2$. We use Lemma 2 to construct $\psi \in \cup_\theta \mathcal{M}_\theta$ such that for any $x, y \in \mathcal{D}$, $|\ell(\psi(x), y) - \ell(h(x), y)| < \Delta/2$. Let $f_1(x, y) = \ell(\psi(x), y)$ and $f_2(x, y) = \ell(1 - y_0, y)$. Denote $f = f_1 - f_2$. Notice that if we assume that the algorithm is consistent on $\cup_\theta \mathcal{M}_\theta$, Lemma 3 gives that $\mathbb{E}f(X, Y) = \frac{1}{r_0} \mathbb{E}f(X, Y) \mathbb{1}_{g(X,Y) \le q^{r_0}}$. We will prove the non-consistency result by means of contradiction, showing that instead we have

$$\mathbb{E}f(X, Y) < \frac{1}{r_0} \mathbb{E}f(X, Y) \mathbb{1}_{g(X,Y) \le q^{r_0}}. \tag{14}$$

To do so, we start by noticing three simple results that are going to be used in the following derivations

- $\forall x \in K_2, y \in \mathcal{Y}, f(x, y) = 0.$

- $\forall x \in K_1, f(x, y_0) = -\ell(1 - y_0, y_0)$ and $f(x, 1 - y_0) = \ell(y_0, 1 - y_0)$

- $\forall x \in \mathcal{B}(x_0, \epsilon) \setminus \mathcal{B}(x_0, \epsilon'), y \in \mathcal{Y}, |f(x, y)| \le \max \ell$

We start be upper bounding the left hand side of (14) as follows:

$$\begin{aligned}
\mathbb{E}f(X, Y) &= \mathbb{E}f(X, Y)\big[\mathbb{1}_{X \in K_1} + \mathbb{1}_{X \in K_2} + \mathbb{1}_{X \in \mathcal{B}(x_0, \epsilon) \setminus \mathcal{B}(x_0, \epsilon')}\big] \\
&\le \mathbb{P}\Big(X \in K_1, Y = 1 - y_0\Big) \ell(y_0, 1 - y_0) \\
&\quad - \mathbb{P}\Big(X \in K_1, Y = y_0\Big) \ell(1 - y_0, y_0) \\
&\quad + \mathbb{P}\Big(X \in \mathcal{B}(x_0, \epsilon) \setminus \mathcal{B}(x_0, \epsilon')\Big) \max \ell \\
&< \mathbb{P}\Big(X \in K_1, Y = 1 - y_0\Big) \ell(y_0, 1 - y_0) + \frac{\Delta}{2}
\end{aligned}$$

Using (13), we can lower bound the right hand side of (14) as follows:

$$\begin{aligned}
\frac{1}{r_0} \mathbb{E}f(X, Y) \mathbb{1}_{g(X,Y) \le q^{r_0}} &= \frac{1}{r_0} \mathbb{E}f(X, Y)\big[\mathbb{1}_{X \in K_1} + \mathbb{1}_{X \in K_2} + \mathbb{1}_{X \in \mathcal{B}(x_0, \epsilon) \setminus \mathcal{B}(x_0, \epsilon')}\big]\mathbb{1}_{g(X,Y) \le q^{r_0}} \\
&\ge \frac{1}{r_0} \mathbb{P}\Big(X \in K_1, Y = 1 - y_0\Big) \ell(y_0, 1 - y_0) \\
&\quad - \frac{1}{r_0} \mathbb{P}\Big(X \in \mathcal{B}(x_0, \epsilon) \setminus \mathcal{B}(x_0, \epsilon')\Big) \max \ell \\
&> \frac{1}{r_0}\Big[\mathbb{P}\Big(X \in K_1, Y = 1 - y_0\Big) \ell(y_0, 1 - y_0) - \frac{\Delta}{2}\Big] \\
&> \mathbb{E}f(X, Y) \\
&\quad + [\frac{1}{r_0} - 1]\mathbb{P}\Big(X \in K_1, Y = 1 - y_0\Big) \ell(y_0, 1 - y_0) \\
&\quad - \frac{1}{2}[\frac{1}{r_0} + 1]\Delta \\
&> \mathbb{E}f(X, Y),
\end{aligned}$$

where the last line comes from the definition of $\Delta$. $\qquad \square$

### A.4 Proof of Theorem 2

Denote $\mathcal{P}_B$ the set of probability distributions on $\mathcal{X} \times \{0, 1\}$, such that the marginal distribution on the input space is continuous (absolutely continuous with respect to the Lebesgue measure on $\mathcal{X}$) and for which

$$p_\pi : x \mapsto \mathbb{P}_\pi(Y = 1 | X = x)$$

is upper semi-continuous. For a probability measure $\pi \in \mathcal{P}_B$, denote $\pi^X$ the marginal distribution on the input. Denote $\gamma$ the function from $[0, 1] \times [0, 1]$ to $\mathbb{R}$ defined by

$$\gamma(p, y) = p\ell(y, 0) + (1 - p)\ell(y, 1).$$

Finally, denote $\mathcal{F}$ the set of continuous functions from $\mathcal{X}$ to $[0, 1]$. We recall the two assumptions made on the loss:

(i) The loss is non-negative and that $\ell(y, y') = 0$ if and only if $y = y'$

(ii) For $p \in [0, 1]$, $y \mapsto \gamma(p, y) = p\ell(y, 1) + (1 - p)\ell(y, 0)$ has a unique minimizer, denoted $y_p^* \in [0, 1]$, that is increasing with $p$.

**Lemma 4** *Consider a loss $\ell$ that satisfies (ii). Then, for any $p \in [0, 1]$ and $\delta > 0$, there exists $\epsilon > 0$ such that for any $y \in \mathcal{Y} = [0, 1]$,*

$$\gamma(p, y) - \gamma(p, y_p^*) \leq \epsilon \implies |y - y_p^*| \leq \delta.$$

**Proof:** Consider $p \in [0, 1]$ and $\eta > 0$. Assume that for any $\epsilon_k = \frac{1}{k+1}$ there exists $y_k \in \mathcal{Y}$ such that $|y - y_p^*| \geq \delta$ and

$$p\ell(y_k, 1) + (1 - p)\ell(y_k, 0) - p\ell(y_p^*, 1) - (1 - p)\ell(y_p^*, 0) \leq \epsilon_k$$

Since $\mathcal{Y}$ is compact, we can assume that the sequence $(y_k)_k$ converges (taking, if needed, a sub-sequence of the original one). Denote $y_\infty$ this limit. Since $\ell$ and $|\cdot|$ are continuous, it comes that $|y_\infty - y_p^*| \geq \delta$ and

$$p\ell(y_\infty, 1) + (1 - p)\ell(y_\infty, 0) - p\ell(y_p^*, 1) - (1 - p)\ell(y_p^*, 0) = 0,$$

contradicting the assumption that $y_p^*$ is unique. $\qquad \square$

**Lemma 5** *If $\psi$ is a measurable map from $\mathcal{X}$ to $[0, 1]$, then there exists a sequence of continuous functions $f_n \in \mathcal{F}$ that converges point-wise to $\psi$ (for the Lebesgue measure)*

**Proof:** This result is a direct consequence of two technical results, the Lusin's Theorem (Theorem 5 in the appendix), and the continuous extension of functions from a compact set (Theorem 6 in the appendix). $\quad \square$

**Lemma 6** *For a distribution $\pi \in \mathcal{P}_B$. define $\psi_\pi^*$ the function from $\mathcal{X}$ to $[0, 1]$ by*

$$\forall x \in \mathcal{X}, \ \psi_\pi^*(x) = y_{p_\pi(x)}^*$$

*is measurable. Besides,*

$$\inf_{f \in \mathcal{F}} \mathbb{E}_\pi \ell(f(X), Y) = \mathbb{E}_\pi \ell(\psi_\pi^*(X), Y)$$

**Proof:** The function from $[0, 1]$ to $[0, 1]$ defined by

$$p \mapsto \mathrm{argmin}_{y \in [0,1]} \gamma(p, y) = y_p^*,$$

is well defined and increasing from assumption (ii) on the loss. It is, therefore, measurable. Since $p_\pi : x \mapsto \mathbb{P}_\pi(Y = 1 | X = x)$ is measurable, we get that $\psi_\pi^*$ is measurable as the composition of two measurable functions. For the second point, notice that by definition of $\psi_\pi^*$, for any $f \in \mathcal{F}$,

$$
\begin{aligned}
\mathbb{E}_\pi \ell(f(X), Y) &= \mathbb{E}_{\pi_X} \mathbb{E}_\pi \Big[ \ell(f(X), Y) \mid X \Big] \\
&\geq \mathbb{E}_{\pi_X} \mathbb{E}_\pi \Big[ \ell(\psi_\pi^*(X), Y) \mid X \Big] \\
&\geq \mathbb{E}_\pi \ell(\psi_\pi^*(X), Y).
\end{aligned}
$$

Using Lemma 5, we can take a sequence of continuous functions $f_n \in \mathcal{F}$ that converge point-wise to $\psi_\pi^*$. We can conclude using the dominated convergence theorem, leveraging that $\ell$ is bounded. $\qquad \square$

**Lemma 7** *Let $\mathcal{A}$ a* `SBPA` *with an adapted score function $g$ that depends on the labels. Then there exists a compression level $r > 0$ and $\varepsilon > 0$ such that for any $f_0 \in \mathcal{F}$, the two following statements exclude each other*

*(i)* $\mathbb{E}_{\nu_r}\ell(f_0(X), Y) - \inf_{f \in \mathcal{F}} \mathbb{E}_{\nu_r}\ell(f(X), Y) \leq \varepsilon$

*(ii)* $\mathbb{E}_\mu \ell(f_0(X), Y) - \inf_{f \in \mathcal{F}} \mathbb{E}_\mu \ell(f(X), Y) \leq \varepsilon$

**Proof:** Since $g$ depends on the labels, we can find $x_0 \in \mathcal{X}$ in the support of $\mu^X$ such that $p_\mu(x_0) = \mathbb{P}_\mu(Y = 1 \mid X = x_0) \in (0, 1)$ and $g(x_0, 0) \neq g(x_0, 1)$. Without loss of generality, we can assume that $g(x_0, 0) < g(x_0, 1)$. Take $r \in (0, 1)$ such that

$$g(x_0, 0) < q^r < g(x_0, 1)$$

By continuity of $g$, we can find a radius $\eta > 0$ such that for any $x$ in the ball $\mathcal{B}_\eta(x_0)$ of center $x_0$ and radius $\eta$, we have that $g(x, 0) < q^r < g(x, 1)$. Besides, since $p_\mu$ is upper semi-continuous, we can assume that $\eta$ is small enough to ensure that for any $x \in \mathcal{B}_\eta(x_0)$,

$$p_\mu(x) < \frac{1 + p_\mu(x_0)}{2} < 1. \tag{15}$$

Therefore, recalling that $\nu_r = \frac{1}{r}\mu_{|A_r}$

- $\mathbb{P}_{\nu_r}(X \in \mathcal{B}_\eta(x_0)) = \frac{1}{r}\mathbb{P}_\mu(X \in \mathcal{B}_\eta(x_0), Y = 0) > 0$ and $\mathbb{P}_{\nu_r}(Y = 1 \mid X \in \mathcal{B}_\eta(x_0)) = 0$.

- $\mathbb{P}_\mu(X \in \mathcal{B}_\eta(x_0)) > 0$ and $\mathbb{P}_\mu(Y = 1 \mid X \in \mathcal{B}_\eta(x_0)) > 0$.

Denote $\Delta = \mathbb{P}_\mu(X \in \mathcal{B}_\eta(x_0), Y = 1) > 0$. Consider the subset $V$ defined by

$$V = \{x \in \mathcal{B}_\eta(x_0) \ s.t. \ p_\mu(x) \geq \frac{\Delta}{2}\}$$

We can derive a lower-bound on $\mu^X(V)$ as follows:

$$
\begin{aligned}
\Delta &= \int_{x \in \mathcal{B}_\eta(x_0)} p(x)\mu^X(dx) \\
&= \int_{x \in \mathcal{B}_\eta(x_0)} p(x)\mathbb{1}_{p(x) < \frac{\Delta}{2}}\mu^X(dx) + \int_{x \in \mathcal{B}_\eta(x_0)} p(x)\mathbb{1}_{p(x) \geq \frac{\Delta}{2}}\mu^X(dx) \\
&\leq \int_{x \in \mathcal{B}_\eta(x_0)} \frac{\Delta}{2}\mu^X(dx) + \int_{x \in V} \mu^X(dx) \\
&\leq \frac{\Delta}{2} + \mu^X(V).
\end{aligned}
$$

The last inequality gives that $\mu^X(V) \geq \Delta/2 > 0$. Moreover, we can lower-bound $\nu_r^X(V)$ using (15) as follows:

$$
\begin{aligned}
\nu_r^X(V) &= \nu_r(V \times \{0\}) \\
&= \frac{1}{r}\mu(V \times \{0\}) \\
&= \frac{1}{r}\int_{x \in V} (1 - p_\mu(x))\mu^X(dx) \\
&\geq \frac{1 - p_\mu(x_0)}{2r}\mu^X(V) \\
&\geq \frac{1 - p_\mu(x_0)}{4r}\Delta \\
&> 0.
\end{aligned}
$$

Therefore, assumptions i) and ii) on the loss give that $\psi^*_{\nu_r}(x) = 0$ and $\psi^*_\mu(x) \geq y^*_{\frac{\Delta}{2}} > 0$ for any $x \in V$. Using Lemma 4, take $\epsilon_1 > 0$ such that

$$\ell(y, 0) \leq \epsilon_1 \implies y \leq \frac{y^*_{\frac{\Delta}{2}}}{3}. \tag{16}$$

In the following, we will show that there exists $\epsilon_2 > 0$ such that for any $p \geq \frac{\Delta}{2}$,

$$y \leq \frac{y^*_{\frac{\Delta}{2}}}{3} \implies \gamma(p, y) - \gamma(p, y^*_p) \geq \epsilon_2 \tag{17}$$

Otherwise, leveraging the compacity of the sets at hand, we can find two converging sequences $p_k \to p_\infty \geq \frac{\Delta}{2}$ and $y_k \to y_\infty \leq \frac{y^*_{\frac{\Delta}{2}}}{3}$ such that

$$\gamma(p_k, y_k) - \min_{y'} \gamma(p_k, y') \leq \frac{1}{k+1}.$$

Since $\gamma$ is uniformly continuous,

$$p \mapsto \min_{y'} \gamma(p, y')$$

is continuous. Taking the limit it comes that

$$\gamma(p_\infty, y_\infty) - \min_{y'} \gamma(p_\infty, y') = 0,$$

and consequently $y_\infty = y^*_{p_\infty}$. Since $p_\infty \geq \frac{\Delta}{2}$,

$$y_\infty = y^*_{p_\infty} \geq y^*_{\frac{\Delta}{2}} > \frac{y^*_{\frac{\Delta}{2}}}{3}$$

reaching a contradiction.

Now, take $\epsilon_1$ and $\epsilon_2$ satisfying (16) and (17) respectively. Put together, we have that for any $p \geq \frac{\Delta}{2}$,

$$\gamma(0, y) - \gamma(0, y^*_0) \leq \epsilon_1 \implies \gamma(p, y) - \gamma(p, y^*_p) \geq \epsilon_2.$$

Using the definition of $V$, it comes that for any function $f_0$ and $x \in V$

$$\gamma(0, f_0(x)) \leq \epsilon_1 \implies \gamma(p_\mu(x), f_0(x)) - \gamma(p_\mu(x), \psi^*_\mu(x)) \geq \epsilon_2 \tag{18}$$

Let $\varepsilon = r \min(\epsilon_1, \epsilon_2) \frac{\nu^X_r(V)}{4} > 0$. Consider $f_0 \in \mathcal{F}$ satisfying

$$\mathbb{E}_{\nu_r} \ell(f_0(X), Y) - \inf_{f \in \mathcal{F}} \mathbb{E}_{\nu_r} \ell(f(X), Y) \leq \varepsilon.$$

We will prove that

$$\mathbb{E}_\mu \ell(f_0(X), Y) - \inf_{f \in \mathcal{F}} \mathbb{E}_\mu \ell(f(X), Y) > \varepsilon$$

to conclude the proof. Denote $U_{f_0}$ is the subset of $V$ such that for any $x \in U_{f_0}$, $\gamma(0, f_0(x)) \leq \frac{2\varepsilon}{\nu^X_r(V)}$. We get that

$$
\begin{aligned}
\varepsilon &\geq \mathbb{E}_{\nu_r} \ell(f_0(X), Y) - \inf_{f \in \mathcal{F}} \mathbb{E}_{\nu_r} \ell(f(X), Y) \\
&\geq \int_{\mathcal{X}} \left[ \gamma(p_{\nu_r}(x), f_0(x)) - \gamma(p_{\nu_r}(x), \psi^*_{\nu_r}(x)) \right] \nu^X_r(dx) \\
&\geq \int_V \gamma(0, f_0(x)) \nu^X_r(dx) \\
&\geq \frac{2\varepsilon}{\nu^X_r(V)} \nu^X_r(V \setminus U_{f_0})
\end{aligned}
$$

Hence we get that $\nu_r^X(U_{f_0}) \geq \frac{\nu_r^X(V)}{2}$. Since $\frac{2\varepsilon}{\nu_r^X(V)} \leq \epsilon_1$, the right hand side of (18) holds. In other words,

$$\forall x \in U_{f_0}, \gamma(p_\mu(x), f_0(x)) - \gamma(p_\mu(x), \psi_\mu^*(x)) \geq \epsilon_2,$$

from which we successively obtain

$$
\begin{aligned}
\mathbb{E}_\mu \ell(f_0(X), Y) - \inf_{f \in \mathcal{F}} \mathbb{E}_\mu \ell(f(X), Y) &= \int_{\mathcal{X}} \left[ \gamma(p_\mu(x), f_0(x)) - \gamma(p_\mu(x), \psi_\mu^*(x)) \right] \mu^X(dx) \\
&\geq \int_{\mathcal{U}_{f_0}} \left[ \gamma(p_\mu(x), f_0(x)) - \gamma(p_\mu(x), \psi_\mu^*(x)) \right] \mu^X(dx) \\
&\geq \mu^X(U_{f_0}) \epsilon_2 \\
&\geq \mu(U_{f_0} \times \{0\}) \epsilon_2 \\
&= r \, \epsilon_2 \, \nu_r^X(U_{f_0}) \\
&\geq r\epsilon_2 \frac{\nu_r^X(V)}{2} \\
&> \varepsilon.
\end{aligned}
$$

$\square$

We can now ready to prove Theorem 2.

**Theorem 2.** *Let $\mathcal{A}$ a SBPA with an adapted score function $g$ that depends on the labels. If $\cup_\theta \mathcal{M}_\theta$ has the universal approximation property and the loss satisfies assumptions (i) and (ii), then there exist two hyper-parameters $\theta_1, \theta_2 \in \Theta$ such that the algorithm is not valid on $\mathcal{W}_{\theta_1} \cup \mathcal{W}_{\theta_2}$.*

**Proof:** Denote $\tilde{\Theta} = \Theta \times \Theta$, and for $\tilde{\theta} = (\theta_1, \theta_2) \in \tilde{\Theta}$, $\mathcal{W}_{\tilde{\theta}} = \mathcal{W}_{\theta_1} \cup \mathcal{W}_{\theta_2}$ and $\mathcal{M}_{\tilde{\theta}} = \mathcal{M}_{\theta_1} \cup \mathcal{M}_{\theta_2}$. We will leverage Proposition 1 and Lemma 7 show that there exist a compression ratio $r \in (0, 1)$ and a hyper-parameter $\tilde{\theta}$ such that

$$\min_{w \in \mathcal{W}_{\tilde{\theta}}^*(r)} \mathcal{L}(w) > \min_{w \in \mathcal{W}_{\tilde{\theta}}} \mathcal{L}(w)$$

which would conclude the proof.

Using Lemma 7, we can find $r$ and $\epsilon > 0$ such that for any continuous function $f_0 \in \mathcal{F}$, the two following propositions exclude each other:

(i) $\mathbb{E}_\mu \ell(f_0(X), Y) - \inf_{f \in \mathcal{F}} \mathbb{E}_\mu \ell(f(X), Y) \leq \epsilon$

(ii) $\mathbb{E}_{\nu_r} \ell(f_0(X), Y) - \inf_{f \in \mathcal{F}} \mathbb{E}_{\nu_r} \ell(f(X), Y) \leq \epsilon$

Since $\cup \mathcal{M}_\theta$ has the universal approximation property, and that $\psi_\mu^*$ and $\psi_{\nu_r}^*$ (defined as in Lemma 6) are measurable, we consecutively use Lemma 5 and Lemma 2 to find $\tilde{\theta} = (\theta_1, \theta_2)$ such that

1. There exists $\psi_1 \in \mathcal{M}_{\theta_1}$ such that $\mathbb{E}_\mu \ell(\psi_1(X), Y) - \mathbb{E}_\mu \ell(\psi_\mu^*(X), Y) \leq \epsilon/2$

2. There exists $\psi_2 \in \mathcal{M}_{\theta_2}$ such that $\mathbb{E}_{\nu_r} \ell(\psi_2(X), Y) - \mathbb{E}_{\nu_r} \ell(\psi_{\nu_r}^*(X), Y) \leq \epsilon/2$

Take $\psi_1, \psi_2 \in \mathcal{M}_{\tilde{\theta}}$ two such functions. Consider any parameter $w \in \mathrm{argmin}_{w \in \mathcal{W}_{\tilde{\theta}}^*(r)} \mathcal{L}(w)$. By definition, it comes that

$$
\begin{aligned}
\mathbb{E}_{\nu_r} \ell(y_{out}(X; w), Y) - \mathbb{E}_{\nu_r} \ell(\psi_{\nu_r}^*(X), Y) &\leq \mathbb{E}_{\nu_r} \ell(\psi_2, Y) - \mathbb{E}_{\nu_r} \ell(\psi_{\nu_r}^*(X), Y) \\
&\leq \epsilon/2
\end{aligned}
$$

Therefore, since Lemma 6 gives that $\inf_{f \in \mathcal{F}} \mathbb{E}_{\nu_r} \ell(f(X), Y) = \mathbb{E}_{\nu_r} \ell(\psi_{\nu_r}^*(X), Y)$, we can conclude that

$$\mathbb{E}_\mu \ell(y_{out}(X; w), Y) - \inf_{f \in \mathcal{F}} \mathbb{E}_\mu \ell(f(X), Y) > \epsilon,$$

from which we deduce that

$$
\begin{aligned}
\mathbb{E}_\mu \ell(y_{out}(X; w), Y) \quad &> \quad \inf_{f \in \mathcal{F}} \mathbb{E}_\mu \ell(f(X), Y) + \epsilon \\
&> \quad \mathbb{E}_\mu \ell(\psi_1(X), Y) + \epsilon/2 \\
&\geq \quad \min_{w' \in \mathcal{W}_{\tilde{\theta}}} \mathcal{L}(w') + \epsilon/2,
\end{aligned}
$$

which gives the desired result. $\qquad\square$

## A.5  Proof of the Corollaries 3 and 4

These two corollaries are a straightforward application of Theorem 1 and Theorem 2 as well as the existing literature on the universal approximation properties of Neural Networks: (Hornik, 1991) and (Kidger and Lyons, 2020). We give the proof of the result for wide neural networks. Consider any number of hidden layers $H \geq 1$ fixed. Denote $\theta = (K, R) \in \mathbb{N} \times \mathbb{R} = \Theta$. (Hornik, 1991) implies that $\cup_{(K,R) \in \Theta} FFNN_{H,K}^\sigma(R)$ has the universal approximation property. Theorem 1 states that one can find a $\theta_0 = (K_0, R_0)$ such that the `SBPA` is not consistent on $FFNN_{H,K_0}^\sigma(R_0)$. Now from the definition of consistency, we get that if a `SBPA` is not consistent on $\mathcal{M}$, then it is not consistent on any superset $\mathcal{M}'$ that contains $\mathcal{M}$. Therefore, we get the non-consistency result by noticing that

$$
FFNN_{H,K_0}^\sigma(R_0) \subset FFNN_{H,K}^\sigma(R),
$$

for any $K \geq K_0$ and $R \geq R_0$. Similarly, Theorem 2 states that there exist $\theta_1 = (K_1, R_1)$ and $\theta_2 = (K_2, R_2)$ such that the model is not valid for any class of model such that

$$
FFNN_{H,K_1}^\sigma(R_1) \cup FFNN_{H,K_2}^\sigma(R_2) \subset \mathcal{M}.
$$

We can conclude noticing that $FFNN_{H,K}^\sigma(R)$ satisfies this condition if $K \geq \max(K_1, K_2)$ and $R \geq \max(R_1, R_2)$.

## A.6  Proof of Theorem 3

For $K \in \mathbb{N}^*$, denote $\mathcal{P}_C^K$ the set of generating processes for $K$-classes classification problems, for which the input $X$ is a continuous random variable (the marginal of the input is dominated by the Lebesgue measure), and the output $Y$ can take one of $K$ values in $\mathcal{Y}$ (the same for all $\pi \in \mathcal{P}_C^K$). Similarly, denote $\mathcal{P}_R$, the set of generating processes for regression problems for which both the input and output distributions are continuous. Let $\mathcal{P}$ be any set of generating processes introduced previously for regression or classification (either $\mathcal{P} = \mathcal{P}_C^K$ for some $K$, or $\mathcal{P} = \mathcal{P}_R$).

Assume that there exist $(x_1, y_1), (x_2, y_2) \in \mathcal{D}$ such that

$$
\operatorname{argmin}_{w \in \mathcal{W}_\theta} \ell(y_{out}(x_1; w), y_1) \cap \operatorname{argmin}_{w \in \mathcal{W}_\theta} \ell(y_{out}(x_2; w), y_2) = \emptyset. \tag{H1}
$$

For any `SBPA` algorithm $\mathcal{A}$ with adapted criterion, we will show that there exists a generating process $\mu \in \mathcal{P}$ for which $\mathcal{A}$ is not valid. More precisely, we will show that there exists $r_0 \in (0,1)$ such that for any compression ratio $r \leq r_0$, there exists a generating process $\mu \in \mathcal{P}$ for which $\mathcal{A}$ is not valid. To do so, we leverage Corollary 2 and prove that for any $r \leq r_0$, there exists $\mu \in \mathcal{P}$, for which $\mathcal{W}_\theta^*(\nu_r) \cap \mathcal{W}_\theta^*(\mu) \neq \emptyset$, i.e.

$$
\exists r_0 \in (0,1), \ \forall r \leq r_0, \exists \mu \in \mathcal{P} \ s.t. \ \forall w_r^* \in \mathcal{W}_\theta^*(\nu_r), \ \mathcal{L}_\mu(w_r^*) > \min_{w \in \mathcal{W}_\theta} \mathcal{L}_\mu(w) \tag{19}
$$

We bring to the reader's attention that $\nu_r = \frac{1}{r}\mu_{|A_r} = \nu_r(\mu)$ depends on $\mu$, and so does the acceptance region $A_r = A_r(\mu)$.

The rigorous proof of Theorem 3 requires careful manipulations of different quantities, but the idea is rather simple. Fig. 13 illustrates the main idea of the proof. We construct a distribution $\mu$ with the majority of the probability mass concentrated around a point where the value of $g$ is not minimal.

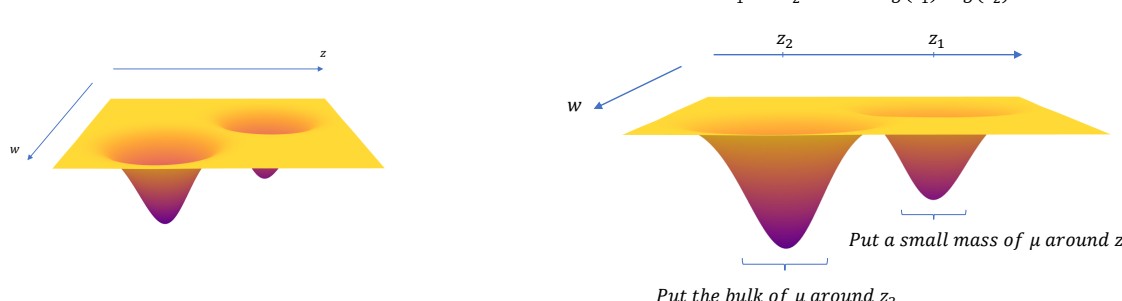

Figure 13: Graphical sketch of the proof of Theorem 3. The surface represents the loss function $f(z,w) = \ell(y_{out}(x), y)$ in 2D, where $z = (x, y)$.

We start by introducing further notations. For $z = (x, y) \in \mathcal{D}$, and $w \in \mathcal{W}_\theta$, we denote by $f$ the function defined by $f(z, w) = \ell(y_{out}(x), y)$. We will use the generic notation $\ell_2$ to refer to the Euclidean norm on the appropriate space. We denote $\mathcal{B}(X, \rho)$ the $\ell_2$ ball with center $X$ and radius $\rho$. If $\mathcal{X}$ is a set, then $\mathcal{B}(\mathcal{X}, \rho) = \bigcup_{X \in \mathcal{X}} \mathcal{B}(X, \rho)$. For $S \subset \mathcal{D}$, we denote $\operatorname{argmin}_w f(\mathcal{S}, w) = \bigcup_{X \in \mathcal{S}} \operatorname{argmin}_w f(X, w)$.

Notice that $f$ is continuous on $\mathcal{D} \times \mathcal{W}_\theta$. Besides, the set data generating processes $\mathcal{P}$ is i) convex and ii) satisfies for all $X_0 \in \mathcal{D}$, $\delta > 0$ and $\gamma < 1$, there exists a probability measure $\mu \in \mathcal{P}$ such that

$$\mu(\mathcal{B}(X_0, \delta)) > \gamma,$$

These conditions play a central role in the construction of a generating process for which the pruning algorithm is not valid. In fact, the non-validity proof applies to any set of generating processes satisfying conditions i) and ii). To ease the reading of the proof, we break it into multiple steps that we list hereafter.

**Steps of the proof:**

1. For all $z_0 \in \mathcal{D}$, the set $\mathcal{W}_{z_0} = \operatorname{argmin}_w f(z_0, w)$ is compact (and non empty).

2. For all $z_0 \in \mathcal{D}, \delta > 0$, there exists $\rho_0 > 0$ such that for all $\rho \leq \rho_0$,

$$\operatorname{argmin}_w f(\mathcal{B}(z_0, \rho), w) \subset \mathcal{B}(\mathcal{W}_{z_0}, \delta)$$

3. Under assumption (H1), there exists $z_1, z_2 \in \mathcal{D}$ such that i) $g(X_1) < g(X_2)$ and ii) $\mathcal{W}_{z_1} \cap \mathcal{W}_{z_2} = \emptyset$

4. For $z_1, z_2$ as in 3, denote $\mathcal{W}_1 = \mathcal{W}_{z_1}$ and $\mathcal{W}_2 = \mathcal{W}_{z_2}$. There exists $\delta, \rho_0 > 0$ such that for any $\rho \leq \rho_0$ and $w_1 \in \mathcal{B}(\mathcal{W}_1, \delta)$, and $w_2^* \in \mathcal{W}_2$

$$\inf_{z \in \mathcal{B}(z_2, \rho)} f(z, w_1) > \sup_{z \in \mathcal{B}(z_2, \rho)} f(z, w_2^*)$$

5. For any $r \in (0, 1)$, there exits a generating process $\mu \in \mathcal{P}$ such that any minimizer of the pruned program $w_r^* \in \mathcal{W}_\theta^*(\nu_r)$ necessarily satisfies $w_r^* \in \mathcal{B}(\mathcal{W}_1, \delta)$ and such that $\mu(\mathcal{B}(z_2, \rho)) \geq 1 - 2r$ for a given $\rho \leq \rho_0$.

6. $\exists r_0 > 0$ such that $\forall r \leq r_0$, $\exists \mu \in \mathcal{P}$ such that $\mathcal{L}_\mu(w_r^*) > \min_{w \in \mathcal{W}_\theta} \mathcal{L}_\mu(w)$ for any $w_r^* \in \mathcal{W}_\theta^*(\nu_r)$

**Proof:** **Result 1:** Let $\mathcal{W}_{z_0} = \operatorname{argmin}_w f(z_0, w) \subset \mathcal{W}_\theta$. Since $\mathcal{W}_\theta$ is compact and functions $f_{z_0} : w \mapsto f(z_0, w)$ is continuous, it comes that $\mathcal{W}_{z_0}$ is well defined, non-empty and closed (as the inverse image of a closed set). Hence it is compact.

**Result 2:** Let $z_0 \in \mathcal{D}$ and $\delta > 0$. We will prove the result by contradiction. Suppose that for any $\rho > 0$, there exists $w \in \operatorname{argmin}_{w'} f(\mathcal{B}(z_0, \rho), w')$ such that $d(w, \mathcal{W}_{z_0}) \geq \delta$.

It is well known that since $f$ is continuous and that $\mathcal{W}_\theta$ is compact, the function

$$z \mapsto \min_{w \in \mathcal{W}_\theta} f(z, w),$$

is continuous. Therefore, for any $k > 0$, we can find $\rho_k > 0$ such that for any $z \in \mathcal{B}(z_0, \rho_k)$,

$$|\inf_w f(z, w) - \inf_w f(z_0, w)| < \frac{1}{k}$$

For every $k > 0$, let $w^k, z^k$ such that $z^k \in \mathcal{B}(z_0, \rho_k)$, $w^k \in \operatorname{argmin}_w f(z^k, w)$ and $d(w^k, \mathcal{W}_{z_0}) \geq \delta$. By definition, $\lim z^k = z_0$. Since $\mathcal{W}_\theta$ is compact, we can assume that $w^k$ converges to $w^\infty$ without loss of generality (taking a sub-sequence of the original one). Now, notice that

$$|f(z^k, w^k) - \inf_w f(z_0, w)| = |\inf_w f(z^k, w) - \inf_w f(z_0, w)| < 1/k,$$

therefore, since $f$ is continuous, $f(z_0, w^\infty) = \inf_w f(z_0, w)$ and so $w^\infty \in \mathcal{W}_{z_0}$, which contradicts the fact that $d(w^k, w^\infty) \geq \delta$ for all $k$. Hence, we can find $\rho > 0$ such that for all $\operatorname{argmin}_w f(\mathcal{B}(z_0, \rho)) \subset \mathcal{B}(\mathcal{W}_{z_0}, \delta)$.

**Result 3:** Let $z_1, z_2$ as in (H1) such that $g(z_1) = g(z_2)$. Since $d$ is continuous, and $\mathcal{W}_1 = \mathcal{W}_{z_1}$ and $\mathcal{W}_2 = \mathcal{W}_{z_2}$ are compact, $d(\mathcal{W}_1 \times \mathcal{W}_2)$ is also compact. Hence, there exists $\delta > 0$ such that

$$\min_{w_1 \in \mathcal{W}_1, \ w_2 \in \mathcal{W}_2} d(w_1, w_2) \geq \delta.$$

Using the previous result, let $\rho$ such that $\operatorname{argmin}_w f(\mathcal{B}(z_1, \rho), w) \subset \mathcal{B}(\mathcal{W}_1, \delta/2)$, The triangular inequality yields $\operatorname{argmin}_w f(\mathcal{B}(z_1, \rho), w) \cap \mathcal{W}_2 = \emptyset$. Since $g$ is adapted and $\mathcal{B}(z_1, \rho)$ has strictly positive Lebesgue measure, we can find $z_1' \in \mathcal{B}(z_1, \rho)$ such that $g(z_1') \neq g(z_1)$. Therefore, the points $z_1', z_2$ satisfy the requirements.

**Result 4:** Since $\mathcal{W}_1$ is compact and $f_{z_2}$ is continuous, $f(z_2, \mathcal{W}_1)$ is compact, and since $\mathcal{W}_1 \cap \mathcal{W}_2 = \emptyset$,

$$\min f(z_2, \mathcal{W}_1) > f(z_2, w_2^*) = \min_{w \in \mathcal{W}_\theta} f(z_2, w),$$

for any $w_2^* \in \mathcal{W}_2$. Denote $\Delta = \min f(z_2, \mathcal{W}_1) - \min_w f(z_2, w) > 0$.

Since $f$ is continuous on the compact space $\mathcal{D} \times \mathcal{W}_\theta$, it is uniformly continuous. We can hence take $\delta > 0$ such that for $z, z' \in \mathcal{D}$ and $w, w' \in \mathcal{W}_\theta$ such that

$$\|z - z'\| \leq \delta, \|w - w'\| \leq \delta \implies |f(z, w) - f(z', w')| \leq \Delta/3.$$

Using Result 2, we can find $\rho_0 > 0$ such that for all $\rho \leq \rho_0$,

$$\operatorname{argmin}_w f(\mathcal{B}(z_1, \rho), w) \subset \operatorname{argmin}_w f(\mathcal{B}(z_1, \rho_0), w) \subset \mathcal{B}(\mathcal{W}_1, \delta)$$

We can assume without loss of generality that $\rho_0 \leq 2\delta$. Let $w_1 \in \mathcal{B}(\mathcal{W}_1, \delta)$. For any $w_2^* \in \mathcal{W}_2$, we conclude that

$$\min_{z \in \mathcal{B}(z_2, \rho)} f(z, w_1) \geq \min f(z_2, \mathcal{W}_1) - \Delta/3 > f(z_2, w_2^*) + \Delta/3 \geq \sup_{z \in \mathcal{B}(z_2, \rho)} f(z, w_2^*).$$

**Result 5:** Let $\rho_0$ defined previously, $k > 1$ and $r \in (0, 1)$. Using the uniform continuity of $f$, we construct $0 < \rho_k \leq \rho_0$ such that

$$\forall w \in \mathcal{P}, \forall z, z' \in \mathcal{D}, d(z, z') \leq \rho_k \implies |f(z, w) - f(z', w)| \leq 1/k.$$

Consider $\mu^k \in \mathcal{P}$ such that $\mu^k\big(\mathcal{B}(z_1,\rho_k)\big) \geq r$ and $\mu^k\big(\mathcal{B}(z_2,\rho_k)\big) \geq 1-r-r/k$. Let $\nu_r^k = \nu_r(\mu^k)$. It comes that $\nu_r^k(\mathcal{B}(z_1,\rho_k)) \geq 1 - \frac{1}{k}$. Using a proof by contradiction, we will show that there exists $k > 1$ such that

$$\arg\min_w \mathbb{E}_{\nu_r^k} f(z,w) \subset \mathcal{B}(\mathcal{W}_1,\delta).$$

Suppose that the result doesn't hold, we can define a sequence of minimizers $w_k$ such that $w_k \in \arg\min_w \mathbb{E}_{\nu_r^k} f(z,w)$ and $d(w_k, \mathcal{W}_1) > \delta$. Denote $M = \sup_{z,w} f(z,w)$. Take any $w_1^* \in \mathcal{W}_1$,

$$
\begin{align}
\mathbb{E}_{\nu_r^k} f(z,w_k) &\leq \mathbb{E}_{\nu_r^k} f(z,w_1^*) \tag{20} \\
&\leq \left( f(z_1,w_1^*) + \frac{1}{k} \right) \nu_k(\mathcal{B}(z_1,\rho_k)) + M\big(1 - \nu_k(\mathcal{B}(z_1,\rho_k))\big) \tag{21} \\
&\leq \left( f(z_1,w_1^*) + \frac{1}{k} \right) + \frac{M}{k} \tag{22} \\
&\leq \left( \min_w f(z_1,w) + \frac{1}{k} \right) + \frac{M}{k} \tag{23}
\end{align}
$$

Similarly, we have that

$$
\begin{align}
\mathbb{E}_{\nu_r^k} f(z,w_k) &\geq \left( f(z_1,w_k) - \frac{1}{k} \right) \nu_k(\mathcal{B}(z_1,\rho_k)) \tag{24} \\
&\geq \left( f(z_1,w_k) - \frac{1}{k} \right)(1 - 1/k) \tag{25} \\
&\geq \left( \min_w f(z_1,w) - \frac{1}{k} \right)(1 - 1/k). \tag{26}
\end{align}
$$

Putting the two inequalities together, we find

$$\left( \min_w f(z_1,w) - \frac{1}{k} \right)(1 - 1/k) \leq \left( f(z_1,w_k) - \frac{1}{k} \right)(1 - 1/k) \leq \left( \min_w f(z_1,w) + \frac{1}{k} \right) + \frac{M}{k}$$

Since $\mathcal{W}_\theta$ is compact, we can assume that $\lim_k w^k = w^\infty \in \mathcal{W}_\theta$ (taking a sub-sequence of the original one). And since $f_{z_1}$ is continuous, we can deduce that $f(z_1, w^\infty) = \min_w f(z_1,w)$, which contradict the fact that $d(w^k, w^\infty) > \delta$ for all $k$.

**Result 6:** Let $r \in (0,1)$ and $\delta, \rho_0, \rho, \mu$ as in the previous results. Let $w_r \in \mathcal{W}_\theta^*(\nu_r)$ From Result 5, we have that $w_r \in \mathcal{B}(\mathcal{W}_1, \delta)$. For $w_2^* \in \mathcal{W}_2$, Result 5 implies that

$$
\begin{aligned}
&\min_{z \in \mathcal{B}(z_2,\rho)} f(z,w_r) - \sup_{z \in \mathcal{B}(z_2,\rho)} f(z,w_2^*) \\
&\geq \min_{z \in \mathcal{B}(z_2,\rho_0)} f(z,w_r) - \sup_{z \in \mathcal{B}(z_2,\rho_0)} f(z,w_2^*) = \Delta \\
&> 0
\end{aligned}
$$

Therefore,

$$
\begin{align}
\mathbb{E}_\mu f(z,w_r) &\geq \min_{z \in \mathcal{B}(z_2,\rho_0)} f(z,w_1) \times \mu(\mathcal{B}(z_2,\rho^r)) \tag{27} \\
&\geq \left( \sup_{z \in \mathcal{B}(z_2,\rho_0)} f(z,w_2^*) + \Delta \right) \mu(\mathcal{B}(z_2,\rho)) \tag{28} \\
&\geq \mathbb{E}_\mu f(z,w_2^*) + \Delta(1 - 2r) - 2rM \tag{29} \\
&\geq \min_w \mathbb{E}_\mu f(z,w) + \Delta(1 - 2r) - 2rM. \tag{30}
\end{align}
$$

Therefore,

$$\mathcal{L}_\mu(w_r) - \min_{w \in \mathcal{W}_\theta} \mathcal{L}_\mu(w) \geq \Delta(1 - 2r) - 2rM,$$

which is strictly positive for $r < \frac{\Delta}{2(M+\Delta)} = r_0$ $\qquad\square$

### A.7 Proof of Proposition 4

**Proposition 4.** [Consistency of Exact Calibration+`SBPA`]
*Let $\mathcal{A}$ be a `SBPA` algorithm. Using the Exact Calibration protocol with signal proportion $\alpha$, the calibrated algorithm $\bar{\mathcal{A}}$ is consistent if $1 - \alpha > 0$, i.e. the exploration budget is not null. Besides, under the same assumption $1 - \alpha > 0$, the calibrated loss is an unbiased estimator of the generalization loss at any finite sample size $n > 0$,*

$$\forall w \in \mathcal{W}_\theta, \ \forall r \in (0,1), \ \mathbb{E}\mathcal{L}_n^{\bar{\mathcal{A}},r,\alpha}(w) = \mathcal{L}(w).$$

**Proof:** Consider $\alpha < 1$. Let $f(z_i, w) = \ell(y_{out}(x_i, w), y_i)$, and $p_e = \frac{(1-\alpha)r}{1-\alpha r}$. For $i \in \{1, ..., n\}$, consider the independent Bernoulli random variables $b_i \sim \mathcal{B}(p_e)$. Notice that

$$\mathcal{L}_n^{\bar{\mathcal{A}},r,\alpha}(w) = \frac{1}{n} \sum_{i=1}^n \left( \mathbb{1}_{z_i \in \mathcal{A}(D_n, \alpha r)} + \frac{b_i}{p_s} \mathbb{1}_{z_i \notin \mathcal{A}(D_n, \alpha r)} \right) f(z_i, w),$$

which gives

$$\mathbb{E}\mathcal{L}_n^{\bar{\mathcal{A}},r,\alpha}(w) = \mathbb{E}_{\mathcal{D}_n} \mathbb{E}\left[ \mathcal{L}_n^{\bar{\mathcal{A}},r,\alpha}(w) \mid \mathcal{D}_n \right] = \mathbb{E}_{\mathcal{D}_n} \mathcal{L}_n(w) = \mathcal{L}(w).$$

Define the random variables

$$Y_{n,i} = \left( \mathbb{1}_{z_i \in \mathcal{A}(D_n, \alpha r)} + \frac{b_i}{p_s} \mathbb{1}_{z_i \notin \mathcal{A}(D_n, \alpha r)} - 1 \right) f(z_i, w),$$

Let $\mathcal{F}_{n,i} = \sigma(\{Y_{n,j}, j \neq i\})$ be the $\sigma$-algebra generated by the random variables $\{Y_{n,j}, j \neq i\}$. Let us now show that the conditions of Theorem 7 hold with this choice of $Y_{n,i}$.

- Let $n \geq 1$ and $i \in \{1, \ldots, n\}$. Similarly to the previous computation, we get that $\mathbb{E}[Y_{n,i} \mid \mathcal{F}_{n,i}] = 0$.

- Using the compactness assumption on the space $\mathcal{W}_\theta$ and $\mathcal{D}$, we trivially have that $\sup_{i,n} \mathbb{E}Y_{n,i}^4 < \infty$.

- Trivially, for each $n \geq 1$, the variables $\{Y_{n,i}\}_{1 \leq i \leq n}$ are identically distributed.

Using Theorem 7 and the standard strong law of large numbers, we have that $n^{-1} \sum_{i=1}^n Y_{n,i} \to 0$ almost surely, and $n^{-1} \sum_{i=1}^n f(z_i, w) \to \mathbb{E}_\mu f(z, w)$ almost surely, which concludes the proof for the consistency.

$\square$

## B Technical results

**Theorem 4 (Universal Approximation Theorem, (Hornik, 1991))** *Let $C(X, Y)$ denote the set of continuous functions from $X$ to $Y$. Let $\phi \in C(\mathbb{R}, \mathbb{R})$. Then, $\phi$ is not polynomial if and only if for every $n, m \in \mathbb{N}$, compact $K \subset \mathbb{R}^n$, $f \in C(K, \mathbb{R}^m)$, $\epsilon > 0$, there exist $k \in \mathbb{N}$, $A \in \mathbb{R}^{k \times n}$, $b \in \mathbb{R}^k$, $C \in \mathbb{R}^{m \times k}$ such that*

$$\sup_{x \in K} \|f(x) - y_{out}(x)\| \leq \epsilon,$$

*where $y_{out}(x) = C^\top \sigma(Ax + b)$.*

**Lemma 8 (Urysohn's lemma, (Arkhangel'skiĭ, 2001))** *For any two disjoint closed sets $A$ and $B$ of a topological space $X$, there exists a real-valued function $f$, continuous at all points, taking the value 0 at all points of $A$, the value 1 at all points of $B$. Moreover, for all $x \in X$, $0 \leq f(x) \leq 1$.*

**Theorem 5 (Lusin's Theorem)** *If $\mathcal{X}$ is a topological measure space endowed with a regular measure $\mu$, if $\mathcal{Y}$ is second-countable and $\psi : \mathcal{X} \to \mathcal{Y}$ is measurable, then for every $\epsilon > 0$ there exists a compact set $K \subset \mathcal{X}$ such that $\mu(\mathcal{X} \setminus K) < \epsilon$ and the restriction of $\psi$ to $K$ is continuous.*

**Theorem 6 (Continuous extension of functions from a compact, (Deimling, 2010))** *Let $A \subset \mathbb{R}^d$ be compact and $f : A \to \mathbb{R}$ be a continuous function. Then there exists a continuous extension $\tilde{f} : \mathbb{R}^d \to \mathbb{R}$ such that $f(x) = \tilde{f}(x)$ for all $x \in A$.*

### B.1 A generalized Law of Large Numbers

There are many extensions of the strong law of large numbers to the case where the random variables have some form of dependence. We prove a strong law of large numbers for specific sequences of arrays that satisfy a conditional zero-mean property.

**Theorem 7** *Let $\{Y_{n,i}, 1 \le i \le n, n \ge 1\}$ be a triangular array of random variables satisfying the following conditions:*

- *For all $n \ge 1$ and $i \in [n]$, $\mathbb{E}[Y_{n,i} \mid \mathcal{F}_{n,i}] = 0$, where $\mathcal{F}_{n,i} = \sigma(\{Y_{n,j}, j \ne i\})$, i.e. the $\sigma$-algebra generated by all the random variables in row $n$ other than $Y_{n,i}$.*

- *For all $n \ge 1$, the random variables $(Y_{n,i})_{1 \le i \le n}$ are identically distributed (but not necessarily independent).*

- $\sup_{n,i} \mathbb{E} Y_{n,i}^4 < \infty$.

*Then, we have that*

$$\frac{1}{n} \sum_{i=1}^{n} Y_{n,i} \to 0, \quad a.s.$$

**Proof:** The proof uses similar techniques to the standard proof of the strong law of large numbers, with some key differences, notably in the use of the Chebychev inequality to upper-bound the fourth moment of the mean. Let $S_n = \sum_{i=1}^{n} Y_{n,i}$. We want to show that $\mathbb{P}(\lim_{n \to \infty} S_n/n = 0) = 1$. This is equivalent to showing that for all $\epsilon > 0$, $\mathbb{P}(S_n > n\epsilon$ for infinitely many $n) = 0$. This event is nothing but the limsup of the events $A_n = \{S_n > n\epsilon\}$. Hence, we can use Borel-Cantelli to conclude if we can show that $\sum_n \mathbb{P}(A_n) < \infty$. Let $\epsilon > 0$. Using Chebychev inequality with degree 4, we have that $\mathbb{P}(A_n) \le (\epsilon n)^{-4} \mathbb{E} S_n^4$. It remains to bound $\mathbb{E} S_n^4$ to conclude. We have that $\mathbb{E} S_n^4 = \mathbb{E} \sum_{1 \le i,j,k,l \le n} Y_{n,i} Y_{n,j} Y_{n,k} Y_{n,l}$. Using the first condition (zero-mean conditional distribution), all the terms of the form $Y_{n,i} Y_{n,j} Y_{n,k} Y_{n,l}$, $Y_{n,i}^2 Y_{n,j} Y_{n,k}$, and $Y_{n,i}^3 Y_{n,l}$ for $i \ne j \ne k \ne l$ vanish and we end up with $\mathbb{E} S_n^4 = n\mathbb{E} Y_{n,1}^4 + 3n(n-1)\mathbb{E} Y_{n,1}^2 Y_{n,2}^2$, where we have used the fact that the number of terms of the form $Y_{n,i}^2 Y_{n,j}^2$ in the sum is given by $\binom{n}{2} \times \binom{4}{2} = \frac{n(n-1)}{2} \times 6 = 3n(n-1)$. Using the last condition of the fourth moment, we obtain that there exists a constant $M > 0$ such that $\mathbb{E} S_n^4 < C n^2$. Using Chebychev inequality, we get that $\mathbb{P}(A_n) \le \epsilon^{-4} n^{-2}$, and thus $\sum_n \mathbb{P}(A_n) < \infty$. We conclude using the Borel-Cantelli lemma. $\qquad\square$

## C Additional Theoretical Results

**Convolutional neural networks:** For an activation function $\sigma$, a real number $R > 0$, and integers $J \ge 1$ and $s \ge 2$ denote $CNN_{J,s}^\sigma(R)$ the set of convolutional neural networks with $J$ filters of length $s$, with all weights and biases in $[-R, R]$. More precisely, for a filter mask $w = (w_0, .., w_{s-1})$, and a vector $x \in \mathbb{R}^d$, the results of the convolution of $w$ and $x$, denoted $w * x$ is a vector in $\mathbb{R}^{d+s}$ defined by $(w * x)_i = \sum_{k=i-s+1}^{i} w_{i-k} x_k$. A network from $CNN_J^\sigma(R)$ is then defined recursively for $x \in \mathcal{X}$:

- $h^{(0)}(x) = x$

- For $j \in [1 : J]$, $h^{(j)}(x) = \sigma(w^{(j)} * h^{(j-1)}(x) + b^{(j)})$, where the filters and biases $w^{(j)}$ and $b^{(j)}$ are in $[-R, R]$

- $y_{out}(x) = c^T h^{(J)}(x)$, where the vector $c$ has entries in $[-R, R]$

**Corollary 5 (Convolutional Neural Networks (Zhou, 2020))** *Let $\sigma$ be the ReLU activation function. Consider a filter length $s \in [2, d_x]$. For any SBPA with adapted score function, there exists a number of filters $J_0$ and a radius $R_0$ such that the algorithm is not consistent on $CNN_{J,s}^{\sigma}(R)$, for any $J \geq J_0$ and $R \geq R_0$. Besides, if the algorithm depends on the labels, then it is also not valid on $CNN_{J,s}^{\sigma}(R)$, for any $J \geq J_0'$ and $R \geq R_0'$.*

# D    Experimental details

| Dataset | CIFAR10 | CIFAR100 |
|---|---|---|
| Architecture | `ResNet18` | `ResNet34` |
| Methods | `GraNd(10)`, `Uncertainty`, `DeepFool` | `GraNd(10)`, `Uncertainty`, `DeepFool` |
| Selection LR | 0.1 | 0.1 |
| Training LR | 0.1 | 0.1 |
| Selection Epochs | 1, 5 | 1, 5 |
| Nb of exps | 3 | 3 |
| Training Epochs | 160 | 160 |
| Optimizer | SGD | SGD |
| Batch Size | 128 | 128 |

The table above contains the different hyper-parameter we used to run the experiments. `GraNd`(10) refers to using the `GraNd` method with 10 different seeds (averaging over 10 different initializations). Selection LR refers to the learning rate used for the coreset selection. The training LR follwos a cosine annealing schedule given by the following:

$$\eta_t = \eta_{min} + \frac{1}{2}(\eta_{max} - \eta_{min})\left(1 + \cos\left(\frac{T_{cur}}{T_{max}}\pi\right)\right),$$

where $T_{cur}$ is the current epoch, $T_{max}$ is the total number of epochs, and $\eta_{max} = 0.1$ and $\eta_{min} = 10^{-4}$. These are the same hyper-parameter choices used by Guo, B. Zhao, and Bai, 2022.

## D.1    MLP for Scaling laws experiments

We consider an MLP given by

$$y_1(x) = \phi(W_1 x_{in} + b_1),$$
$$y_2(x) = \phi(W_2 y_1(x) + b_2),$$
$$y_{out}(x) = W_{out} y_2(x) + b_{out},$$

where $x_{in} \in \mathbb{R}^{1000}$ is the input, $W_1 \in \mathbb{R}^{128 \times 1000}, W_2 \in \mathbb{R}^{128 \times 128}, W_{out} \in \mathbb{R}^{2 \times 128}$ are the weight matrices and $b_1, b_2, b_{out}$ are the bias vectors.

