# OpenReview forum: "Data pruning and neural scaling laws: fundamental limitations of score-based algorithms"
_TMLR — Accepted by TMLR_

### Review · Reviewer_TwLS · 2023-07-28

**Summary Of Contributions:**

The paper focuses on score-based data pruning algorithms (SBPA), which use a scoring function $g$ to rank the examples in the training sample and select a fraction $r$ of the examples accordingly (with the convention that lower values of $g$ point to more important examples).

The authors present some theoretical results, which establish the limitations of score-based pruning. One result (Theorem 1) states that for any SBPA that does not use the labels $y$, suppose we have a collection of function classes $W_\theta$ parameterized by some hyperparameter $\theta$ (e.g. $\theta$ can be the number of layers in a neural network while $W_\theta$ is the set of all functions realized by the neural network with depth $\theta$). If $\cup_\theta W_\theta$ enjoys the universal approximation property, as in deep neural networks, then there exists a hyperparameter $\theta$ such that pruning is inconsistent with respect to the function class $W_\theta$. That is, there exists a function $f\in W_\theta$ such that the expectation of the loss w.r.t. pruned data is different from the expectation of the loss on the original distribution. Hence, model selection based on pruned data can fail.

The intuition behind this result is quite simple. Since it's assumed that SBPA does not use the labels, score-based pruning is asymptotically a rejection sampling method on the instance space $X$ so the pruned data have limiting distribution $\eta$. One can then construct many functions that agree on the support of $\eta$ and disagree outside it, but the scoring rule cannot distinguish among them.

The above result is not that interesting because most score based pruning methods use the labels $y$. This brings us to the second main result in the paper (Theorem 2) , which is more interesting (but I have to admit that I could not really follow its proof). It states that for any SBPA that also uses the labels $y$, there exists a compression level $r_0$ and hyperparameters such that pruning is not "valid" (the paper refers to the classical notion of “consistency” as “validity”).

Initially, I found this last result surprising because the claim is that this holds for any data distribution $\mu$! However, it seems now to be less exciting than I originally thought. This is because the theorem states that there exists *some* hyperparameter $\theta$ where failure happens. In the case of deep neural networks, for example, we do not know if that $\theta$ corresponds to something unrealistic like 1 billion layers.

Generally, the intuition behind such arguments is that pruning changes the data distribution. So, to fix this problem, the authors propose using a mixture of two distributions: pruned data using SBPA and random sampling. Since score-based pruning can be viewed as rejection sampling, one can assign different weights to the two distributions such that we recover the original objective function. The authors show that with as little as 10% of random examples, the consistency is recovered in practice. In theory, of course, any fraction is enough as long as it is positive.

Overall, this is a useful message. However, it does not mean that one can do better than random pruning, which I find to be disappointing. The authors hint that one may do better depending on the quality of SBPA, e.g. with linear datasets, but I don’t find the empirical results convincing. There is no improvement in Figure 11 while Figure 12 doesn’t include a confidence interval. In general, if there is indeed an improvement, it seems to be quite marginal at best.


**Audience:**

Yes

**Claims And Evidence:**

Yes

**Requested Changes:**

- Please include random pruning in Section 5.1, particularly in Figures 6 and 7. Please fix the issue mentioned above about calling r=1 random pruning.
- In Figure 12, please describe how $\alpha$ is selected and whether or not the same $\alpha$ is used for all compression levels $r$. Also, it would be useful to provide confidence intervals for the “best $\alpha$” curves.

Nitpicks:
- Page 4: “euclidiean” → “Euclidean”
- The term “validity” in Definition 2 is used to describe what is often referred to in the literature as “consistency.” This can be confusing.
- Figure 4 does not add much value in my opinion.
- Page 14: there are many extra spaces before the footnote.


**Strengths And Weaknesses:**

*Strengths*

- The paper is well-written overall and provides an excellent theoretical account of the limitations of score-based data pruning methods. It brings to attention their pitfalls and why one might need to supplement them with random sampling to remove the bias introduced by the pruning procedure.

- Some of the examples and discussions in the paper, such as in Section 5.1 using logistic regression, are quite insightful and provide a strong demonstration of the limitation of SBPA algorithms, (in particular GraNd).


*Weaknesses*

- The primary concern in my opinion is in Section 5. The authors state that $r=1$ corresponds to random pruning when, in fact, $r=1$ corresponds to no pruning at all. This is a concern because many of the conclusions in that section pertain to the comparison with random pruning, but there is no comparison with random pruning as far I as understand. In Figures 6 and 7, random pruning should be included for all values of $r$.  Later in Figure 10, the authors correct this when they correctly refer to $r=1$ as “no pruning” but do not fix this in the discussion.

- In many places, the statements of the theorems/propositions are not self-contained. For example, Theorem 1 does not say that it only addresses SBPA algorithms which ignore the labels $y$, even though this becomes evident from the proof and from the subsequent discussion in the paper. Also, in Proposition 5, the functions $f_1$ and $f_2$ are assumed to be in $\mathcal{M}$ but this is not mentioned in the statement of the proposition.
- Along similar lines, I find it surprising that there does not seem to be a smoothness assumption made on the scoring function $g$. This might be embedded within the proof; I don't know. The reason I find this surprising is because one can “approximate” random pruning with a deterministic scoring function $g$ that partitions the domain into extremely small regions, such that pruning based on $g$ is almost similar to random pruning. This is because the probability measure of isolated points is assumed to be zero. However, $g$ would not be smooth in that case, so I would expect smoothness of $g$ to show up as a condition in the results.
- See my comment above about the limitation of some of the theoretical results.
- Figure 12 is supposed to be the main demonstration that one can do better than random pruning by carefully selecting the parameter $\alpha$. But, the authors do not explain how $\alpha$ is selected in that figure and whether or not the same $\alpha$ is used for all compression levels $r$. There are no confidence intervals and the improvement compared to random pruning, if it exists, is quite small.

---

### Review · Reviewer_oDiF · 2023-08-14

**Summary Of Contributions:**

The paper studies score-based dataset running approaches focusing on a setup where less than 30% data is kept. The paper shows that in such setup score-based approaches underperform random pruning. The paper shows how to improve score-based pruning by mixing it with random sampling in high data pruning scenarios. This paper is mostly focused on theory, however, quite broad experimental validation is also included (including CIFAR10 and CIFAR100 datasets).


**Audience:**

Yes

**Broader Impact Concerns:**

The paper covers some limitations of the framework.

I do not have any further suggestions to include.

**Claims And Evidence:**

Yes

**Requested Changes:**

Fig 2 is a bit confusing. For the bottom plots, the legend is not always matching the plots. W, r, and n are not defined in the caption.

I’m curious about the time required by pruning algorithms vs time required for training the classification model. I would assume that that in some cases the time spend on data pruning might not be worth it as it exceeds the time spend on training, the opposite is probably also the case in many scenarios. Adding some discussion about the tradeoff of the time spend on pruning vs time spend on the training of the downstream task model might be interesting.


**Strengths And Weaknesses:**

In general, I liked the paper, one can see that the authors put a lot of effort into the preparation of the manuscript.

Strengths:
- The paper is well written and easy to follow.
- The claims of the paper are well supported in the manuscript.
- The developed theory is easy to follow.
- The review of related works looks complete.


Weaknesses:
- The paper is a bit long. Notation paragraph appears multiple times at different places in the manuscript.  In the Intro, in order to understand some figures, one needs to read sections that are further down the manuscript — as such the flow of the reading is affected. Spending some time in trimming the paper and improving the presentation outline would improve the paper.

- Showing the results on other tasks that go beyond classification and trying to highlight the utility of the developed framework for larger scale datasets such as ImageNet, would further improve the manuscript.

- The observations/conclusions are not surprising, it not an expectation from the TMLR paper to show surprising results, thus this point is  not affecting my judgement of the paper. However, I do think that papers with surprising/interesting observations, results will be more impactful in the long run. Thus, if possible I would encourage the authors to think if there is some angle on the paper presentation and results that would add a twist of surprise to the paper.

---

### Review · Reviewer_moww · 2023-08-27

**Summary Of Contributions:**

1. An interesting novel formalism to characterize the asymptotic properties of data pruning algorithms.
2. Analyses of Scire-Based Pruning Algorithms (SBPA) for universal approximations and general function classes. Theoretical results show that SBPA is worse than random sampling.
3. New algorithm that interpolates between SBPA and random sampling. Theoretical and empirical analyses show that the new method is promising.
4. Empirical evaluations that verify the theories.

**Audience:**

Yes

**Broader Impact Concerns:**

No ethical concerns.

**Claims And Evidence:**

No

**Requested Changes:**

See weaknesses.

**Strengths And Weaknesses:**

Strengths:
1. This paper studies a very relevant problem. I really appreciate the proposed formalism to study this problem in the asymptotic regime.
2. The theoretical results on SPBA are very interesting and, to my knowledge, are new in the literature, although I have some questions regarding their applications to neural networks.
3. The new algorithm is quite straightforward but seems very effective in practice.
4. The empirical verifications are solid.

Weaknesses:
1. I have some questions about corollaries 3 & 4 and in general, about the results of universal approximation. Note the universal approximation results are also asymptotic in nature. For example, the wide neural network requires the width to go to infinity. For fixed $\epsilon$, there exists $R_0$ and $K_0$ in Corollary 3, but I think Corollary 3 is not for fixed $\epsilon$ but letting $\epsilon \rightarrow 0$ as well. I recommend writing Corollarys 3 & 4 and their proofs in a fully rigorous form instead of just citing the universal approximation theorems.
2. While it is definitely valuable to study the asymptotic regime (number of parameters fixed, $n \rightarrow \infty $), it is also worth studying both the number of parameters and $n$ going to infinity (with the same or different rates). I know this is beyond the scope of this paper, but more discussions are needed, because this is a practically relevant scenario.

---

### Author Response · Authors · 2023-08-31
**General Response**

We thank all the reviewers for their positive and constructive feedback. To address their concerns, we have made the following changes in the revised version of the paper (already uploaded to OpenReview):

1. Clarification of the proof of Thm1
2. Clarification of the case for random pruning
3. Merged notation paragraphs
4. Added definitions of $r,n,m,w$ in Fig2
5. Added discussion about pruning time Vs training time
6. Added more detailed proofs of Corollaries 3 and 4
7. Added discussion about proportional limit


Regards,

The authors,

---

### Public Comment · ~Muyang_He1 · 2024-02-28
**Question about Proof of Proposition 3 (Page 26)**

There may exist a mistake in the proof of Propostion 3 in Page 26, the third "$\le$" in line 5. The "$\le$" is doubtful due to $ |a|+|b| \ge |a+b|$ , i.e. $$\sum |a_i| \ge |\sum a_i| $$, which means the equation of 4th line is greater than or equal to that of 5th line ($|G_n(q_n^r)-G_n(q^r)|$). Hence, the proof process is problematic.

$ \frac{1}{n}\sum  _{z\in D  _n}|\mathbb{I} _{g(z)\leq q _n^r}-\mathbb{I} _{g(z)\leq q^r}|=\sum _{z\in D _n}|\frac{1}{n}\mathbb{I} _{g(z)\leq q _n^r}-\frac{1}{n}\mathbb{I} _{g(z)\leq q^r}| \geq |\sum _{z\in D _n}\frac{1}{n}\mathbb{I} _{g(z)\leq q _n^r}-\sum _{z\in D _n}\frac{1}{n}\mathbb{I} _{g(z)\leq q^r}|  =|G _n(q _n^r)-G _n(q^r)| $

---

> ### Author Response · Authors · 2024-02-28
> **Clarification**
>
> Dear Muyang He,
>
> We appreciate your interest in our work. The proof is correct. However we should have used "=" instead of "$\leq$," which can be confusing due to the triangular inequality, as you pointed out.
>
> Indeed, assume that $q^r \leq q_n^r$, then *for any $z$*, $\mathbb{I}\left(g(z) \leq q^r_n \right) \geq \mathbb{I}\left( g(z) \leq q^r \right)$, successively giving
> $$ \frac{1}{n} \sum_{z \in D_n} \left| \mathbb{I} \left( g(z) \leq q^r_n \right) - \mathbb{I} \left(g(z) \leq q^r\right) \right| =  \frac{1}{n} \sum_{z \in D_n} \left( \mathbb{I}\left( g(z) \leq q^r_n \right) - \mathbb{I}\left( g(z) \leq q^r \right) \right) = G_n(q^r_n)-G_n(q^r) = \left| G_n(q^r_n)-G_n(q^r) \right|$$
>
> Similarly, if $q^r \geq q_n^r$, then *for any $z$*, $\mathbb{I}\left(g(z) \leq q^r_n \right) \leq \mathbb{I}\left( g(z) \leq q^r \right)$, giving that the quantity of interest is equal to
> $G_n(q^r)-G_n(q^r_n) = \left| G_n(q^r_n)-G_n(q^r) \right|$
>
> Please do not hesitate to let us know if we overlooked some details or if you find something else unclear.
>
> Best,
>
> The authors

---

### Decision · Action_Editors · 2023-10-27

**Recommendation:** Accept as is

**Comment:**

I'm recommending acceptance based in part on the unanimous agreement among the reviewers that this paper is acceptable and that the paper's claims are well-supported by evidence (btoh empirical and theoretical) The reviewers highlighted the paper's valuable contribution to data pruning and its likely significance as a baseline for future works in the area. I agree with this second point very much. The paper is moreover on an important topic, of considerable interest to the community. Based on my own reading, it is a nice contribution to this line of work.

The authors were very responsive and diligent in revising their manuscript, and addressed a good number of concerns raised by reviewer. Specifically, they have clarified the proof of the main theorem and improved the overall readability of their paper by refining the notation and adding/clarifying key notions.

**Audience:**

Pruning is an active area of research and one of great interest also to practicioners interesting in accelerating training and/or reducing memory usage. The questions answered here also get at important scientific questions about the behavior of deep learning and explanations for its successes and failures. I think the article is clearly above the bar in terms of audience.

**Claims And Evidence:**

The authors have actively addressed the concerns raised during the review process. They've clarified the proofs supporting their theorems and added depth to existing proofs, thereby strengthening the evidence base for their claims. One reviewer found that the experimental results solidly back the paper's claims. Another felt that their previous concerns were sufficiently addressed, indicating that the authors successfully filled any evidence gaps. Given also the unanimous reviewer recommendations and the authors' diligent revisions, these criterion are met.